# KeySync: A Robust Approach for Leakage-free Lip Synchronization in High Resolution

**Antoni Bigata**                                                   *antoni.bigatacasademunt@natwest.com*
*Natwest*

**Rodrigo Mira**                                                              *rs2517@imperial.ac.uk*
*Imperial College London*

**Stella Bounareli**                                                   *stella.bounareli@natwest.com*
*Natwest*

**Michał Stypułkowski**                                        *michal.stypulkowski@cs.uni.wroc.pl*
*University of Wrocław*

**Konstantinos Vougioukas**                              *konstantinos.vougioukas@natwest.com*
*Natwest*

**Stavros Petridis**                                                 *stavros.petridis@natwest.com*
*Natwest*

**Maja Pantic**                                                         *maja.pantic@natwest.com*
*Natwest*

**Reviewed on OpenReview:** *https://openreview.net/forum?id=dvtMHhZUyG*

## Abstract

Lip synchronization, known as the task of aligning lip movements in an existing video with new input audio, is typically framed as a simpler variant of audio-driven facial animation. However, as well as suffering from the usual issues in talking head generation (e.g., temporal consistency), lip synchronization presents significant new challenges such as expression leakage from the input video and facial occlusions, which can severely impact real-world applications like automated dubbing, but are largely neglected by existing works. To address these shortcomings, we present KeySync, a two-stage framework that succeeds in mitigating the issue of temporal consistency, while also incorporating solutions for leakage and occlusions using a carefully designed masking strategy. We show that KeySync achieves state-of-the-art results in lip reconstruction and cross-synchronization, improving visual quality and reducing expression leakage according to LipLeak, our novel leakage metric. Furthermore, we demonstrate the effectiveness of our new masking approach in handling occlusions and validate our architectural choices through several ablation studies. Our code and videos are available here: https://antonibigata.github.io/KeySync/.

## 1 Introduction

Audio-driven facial animation has recently seen substantial progress with the introduction of new generative models such as Generative Adversarial Networks (GANs) Goodfellow et al. (2020); Vougioukas et al. (2019); Zhou et al. (2019) and diffusion models Ho et al. (2020); Stypulkowski et al. (2024); Chen et al. (2024b). In contrast, the adjacent field of lip synchronization (also known as lip-sync) has experienced comparatively slower advancements Guan et al. (2023); Zhang et al. (2023d); Prajwal et al. (2020). This disparity is surprising given that lip-sync has similar applications, ranging from facilitating multilingual content production to

enhancing virtual avatars Zhen et al. (2023); Zhan et al. (2023). A potential reason for this slower progress is that while lip synchronization may seem like a simpler task than animating the full face from audio, it presents unique challenges that remain largely unaddressed.

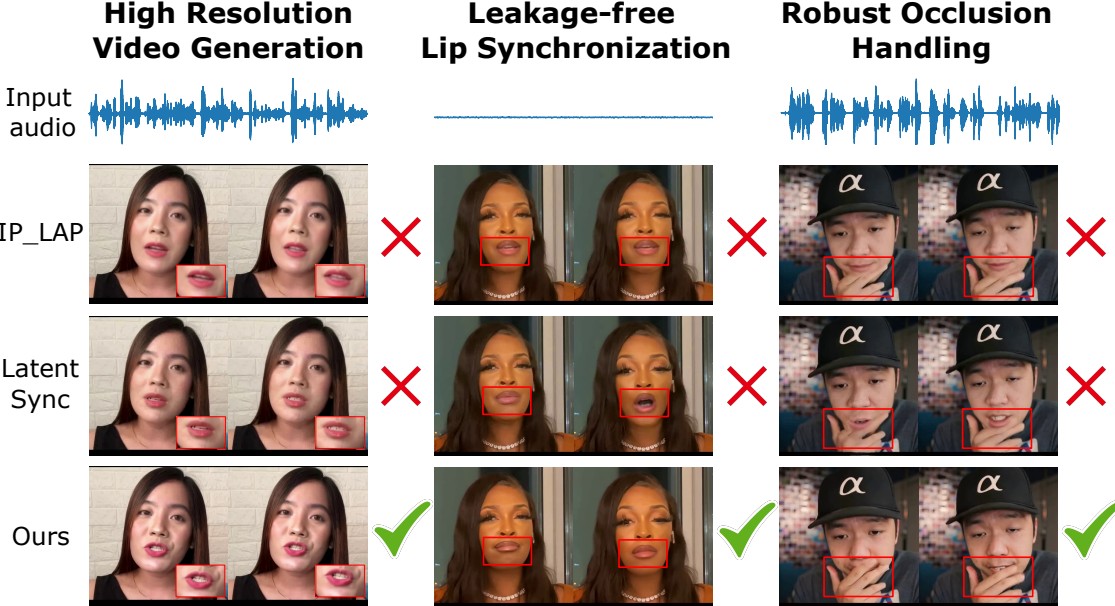

Figure 1: Unlike existing methods, KeySync generates high-resolution lip-synced videos that are closely aligned with the driving audio while minimizing leakage from the input video and seamlessly handling facial occlusions.

Current methods are limited in both visual quality and temporal consistency. While recent works like LatentSync Li et al. (2024) have begun to explore higher resolutions (512x512), many models remain constrained to 256x256 outputs, hindering real-world applicability. Furthermore, they struggle with temporal stability; frame-based approaches Yu et al. (2024); Liu et al. (2024) often produce visible discontinuities, while attempts to enforce coherence indirectly through perceptual models Li et al. (2024), sequence discriminators Mukhopadhyay et al. (2024), or autoregressive conditioning Bigioi et al. (2024) can introduce subtle artifacts or suffer from error accumulation over long sequences.

Beyond temporal consistency, a key, but often overlooked issue is expression leakage, where models infer mouth shapes from facial expressions in the source video rather than from the driving audio. Regrettably, most existing works focus excessively on lip synchronization as a reconstruction task on paired audio-visual data, and neglect the cross-synchronization scenario, where a non-matching audio clip is used to re-animate the original video. As a consequence, they typically exhibit major expression leakage from the original video, severely degrading the synchronization between the generated video and the input audio in the latter scenario. Notably, this behaviour jeopardizes the viability of these models for applications like automated dubbing, where audio and video are naturally mismatched.

To alleviate expression leakage, some methods Cheng et al. (2022); Yaman et al. (2024) introduce an additional network to generate a neutral version of the input frame, neglecting the underlying issue of the masking strategy. Some methods mask only the mouth region while preserving facial areas such as the jaw and cheeks from the original videos, potentially leading to leakage since these regions also convey information about mouth movements Ki & Min (2023); Zhang et al. (2023d), while others adopt broader masks that risk discarding important contextual cues Zhang et al. (2024); Cheng et al. (2022). Remarkably, the impact of these masking strategies on generalization and robustness remains largely unexplored, and no consensus exists on the optimal approach. Lastly, another potential complication lies in occlusion handling. Most existing models assume an unobstructed view of the mouth, whereas, in the real world, occlusions caused by hands,

objects, or motion blur are frequent. In practice, this means that the lack of explicit occlusion-handling mechanisms significantly limits the applicability of current models.

To address these challenges, we propose KeySync, a two-stage lip synchronization framework that leverages recent advances in facial animation to generate high-fidelity videos with lip movements that are temporally consistent and aligned with the input audio. To minimize leakage from the input video, we devise a masking strategy that adequately covers the lower face while retaining the necessary contextual regions. Furthermore, we augment this mask by excluding facial occlusions using a video segmentation model, resulting in a method that consistently handle occlusions without uncanny visual hallucinations. Our primary contributions, illustrated in Figure 1, are:

- **State-of-the-art lip synchronization**: KeySync achieves state-of-the-art lip synchronization performance at a resolution of $(512 \times 512)$, surpassing the common $(256 \times 256)$ standard. It outperforms all competing methods in terms of quality and lip movement accuracy according to several objective metrics and a holistic user study. We observe particularly noticeable improvements in the cross-synchronization setting (where there is a mismatch between the input video and audio), enabling promising real-world applications such as automated dubbing.

- **A new strategy for occlusion handling**: We propose an inference-time strategy for occlusion handling by excluding occluding objects from our mask automatically using a pre-trained video segmentation model. Through qualitative and quantitative analysis, we show this method is consistently effective in handling occlusions.

- **A novel leakage metric**: We propose LipLeak, the first metric to quantify lip synchronization leakage. It measures how much motion from the source video leaks into the output by computing the ratio in lip activity between videos generated using speech versus silent audio.

## 1.1 Claims and Target Audience

In this paper, we claim that expression leakage and poor occlusion handling are critical bottlenecks in high-resolution lip synchronization, and we provide evidence that our proposed masking strategy and two-stage diffusion framework effectively mitigate these issues. We further claim that our novel LipLeak metric successfully quantifies expression leakage, supported by correlation with human perception. This work will be of particular interest to audience working on generative models, especially those focused on video editing, automated dubbing, and the practical deployment of audio-driven facial animation.

## 2 Related Works

**Audio-Driven Facial Animation** Audio-driven facial animation methods aim to generate realistic talking head videos with accurate lip-sync and preserved identity. Early GAN-based works Vougioukas et al. (2019); Zhou et al. (2019); Chung et al. (2017) focused on lip-sync, while later approaches incorporated head pose modelling but often introduced artifacts and unnatural motion Chen et al. (2020); Zhang et al. (2023c); Zhou et al. (2021).

Diffusion models Ho et al. (2020); Rombach et al. (2022) have since emerged as a superior alternative, demonstrating improved temporal consistency and video quality Xu et al. (2024b). Several modern methods leverage video diffusion models for temporally consistent motion Stypulkowski et al. (2024); Xu et al. (2024a). Others condition the generation process on facial landmarks Wei et al. (2024) or 3D meshes Zhang et al. (2023a); however, these approaches often produce unrealistic facial motion. To improve identity reconstruction, recent works Chen et al. (2024b); Xu et al. (2024a) leverages ReferenceNet Hu (2024), though at the cost of increased computational complexity. However, these state-of-the-art methods, including recent keyframe-based techniques Bigata et al. (2025), are designed for full-face generation. Our work addresses the distinct challenge of lip-sync editing, which involves unique problems such as expression leakage from the source video.

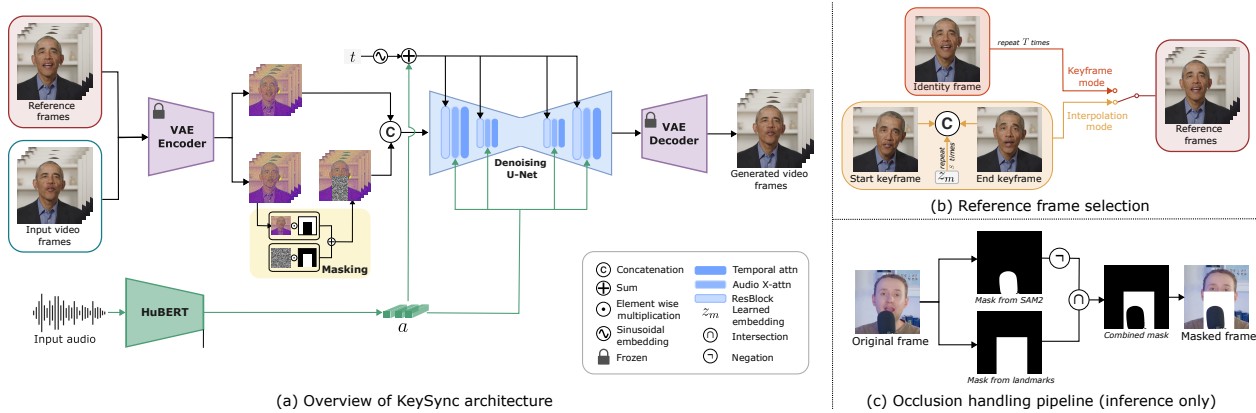

Figure 2: Overview of the KeySync framework. This two-stage latent diffusion model conditions on audio ($a$) and an input video. (b) The keyframe stage uses an identity frame $z_{id}$, while the interpolation stage uses keyframes ($z_i, z_{i+1}$) and intermediate embeddings ($z_m$). (c) Our inference-time occlusion handling pipeline.

**Audio-Driven Lip Synchronization** Lip synchronization methods focus on adjusting mouth movements to match an audio input while preserving other facial attributes, such as head pose and upper face expressions. A foundational work, Wav2Lip Prajwal et al. (2020), uses a Generative Adversarial Network (GAN) to generate lip-synced frames, leveraging a pre-trained expert model to ensure accuracy. To enhance realism and identity generalization, subsequent methods have introduced StyleGAN2-based architectures Guan et al. (2023); Ki & Min (2023), spatial deformation of feature maps Zhang et al. (2023d), and coarse-to-fine pyramid models Muaz et al. (2023). Other approaches include LipFormer Wang et al. (2023b), which uses a codebook of face parts aligned with the audio, and TalkLip Zhong et al. (2023), which employs contrastive learning to improve the quality of the generated lip region. More recently, diffusion-based methods have been introduced for lip synchronization Mukhopadhyay et al. (2024); Liu et al. (2024); Bigioi et al. (2024), marking a shift in the state-of-the-art.

Despite these advances, several key challenges remain. The first is expression leakage, which is particularly problematic in cross-driving scenarios where one person's expression is transferred to another. This leakage often stems from suboptimal masking strategies that fail to cover all visual cues of speech. While some methods Cheng et al. (2022); Yaman et al. (2024) address this by neutralizing the source face, this approach adds computational overhead and potential errors from the synthetic input. To date, no consensus exists on an optimal masking strategy.

A second challenge is temporal consistency. Many methods Yu et al. (2024); Liu et al. (2024); Zhong et al. (2024) operate on a frame-by-frame basis, leading to visible discontinuities. Models that condition on past frames Bigioi et al. (2024) can suffer from cumulative error propagation, while other techniques like perceptual models Li et al. (2024) or sequence discriminators Mukhopadhyay et al. (2024) are often insufficient to guarantee coherence.

Finally, occlusion handling remains a largely unsolved problem. Most models assume an unobstructed view of the mouth, failing in real-world settings with occlusions from hands, objects, or motion blur. Notably, Peng et al. (2025) propose a mask-free method lip sync method, which succeeds in handling occlusions, but falls short in terms of lip synchronization.

## 3 Method

In this section, we describe our two-stage lip-sync approach, followed by our masking strategy in Section 3.2 and a new method for handling occlusions in Section 3.5.

### 3.1 Latent Diffusion

Diffusion models Ho et al. (2020); Dhariwal & Nichol (2021) progressively transform random noise into structured data by iteratively removing noise through a learned denoising process. Latent diffusion Rombach et al. (2022) applies this denoising operation in a compressed latent space rather than in the high-dimensional pixel space, improving computational efficiency. Furthermore, the EDM framework Karras et al. (2022) defines the denoising operation of the denoiser $D_\theta$ as:

$$D_\theta(\mathbf{x}; \sigma) = c_{\text{skip}}(\sigma)\mathbf{x} + c_{\text{out}}(\sigma)F_\theta(c_{\text{in}}(\sigma)\mathbf{x}; c_{\text{noise}}(\sigma)), \tag{1}$$

where $F_\theta$ is the trainable neural network and $\mathbf{x}$ the input. The terms $c_{\text{noise}}(\sigma)$, $c_{\text{out}}(\sigma)$, $c_{\text{skip}}(\sigma)$, and $c_{\text{in}}(\sigma)$ are scaling factors dependent on the noise level $\sigma$. These scaling factors dynamically adjust the magnitude and influence of noise at different stages of the denoising process, thereby improving the network's efficiency and robustness during diffusion.

### 3.2 Leakage-Proof Masking

We frame the lip-sync task as a video inpainting problem Quan et al. (2024); Saharia et al. (2022) in the latent space. The critical objective is to ensure the newly generated lip region does not reuse (or "leak") cues from the original mouth shape that contradict the new audio. Specifically, we create a mask $M$ by computing facial landmarks Bulat & Tzimiropoulos (2017) and isolating the lower facial region, extending slightly above the nose to cover any upper cheek movements that could otherwise convey information about lip movements, while still preserving overall facial identity. The mask also extends to the lower edge of the image, preventing any leakage from jaw movements. We find that this mask strikes an appropriate balance between the two types of masks presented in prior works, namely:

- **Full lower-face masks** Shen et al. (2023); Mukhopadhyay et al. (2024); Park et al. (2022), which can obscure too much context, risking issues with identity and natural facial continuity;

- **Mouth-only masks** Zhang et al. (2023d); Liu et al. (2024); Ki & Min (2023), which can inadvertently leak lower face expressions because residual mouth movements or shading remain visible to the model.

While we group prior work into these two categories, it's important to note that each method implements its own masking strategy, and the exact details are not always shared. This highlights the need for a standardized approach. We provide pseudocode to reproduce our mask and a deeper discussion on the topic in Appendix D. Additionally, a visualization of our approach is available in Figure 6, with baseline comparisons shown in Figure 9.

### 3.3 Two-Stage Video Generation

Our approach is illustrated in Figure 2 and detailed in Algorithm 1. We adapt the two-stage procedure of KeyFace Bigata et al. (2025) to the task of audio-driven video inpainting. We feed the video frames $\{x_t\}_{t=1}^T$ into our VAE encoder Blattmann et al. (2023) $\mathcal{V}$ to obtain latent representations $\{z_t\}_{t=1}^T$. We then add noise to obtain their corresponding noisy versions $\{z_t^n\}_{t=1}^T$. To enforce consistency with the unmasked face (e.g., eyes, cheeks), we formulate the input to the U-Net using a binary mask $M$ (where 0 is the face and 1 is the mouth region):

$$z_t^{in} = M \odot z_t^n + (1 - M) \odot z_t, \tag{2}$$

where $\odot$ denotes element-wise multiplication. This formulation forces the model to reconstruct (inpaint) only the masked mouth region, ensuring the generated lip movements blend seamlessly with the original video.

We aim to generate video frames $\{\hat{x}_t\}_{t=1}^T$ where lip movements are synchronized with a given audio track $\{a_t\}_{t=1}^T$. Unlike previous approaches that either generate all frames end-to-end Ki & Min (2023); Wang et al. (2023a); Li et al. (2024) or explicitly disentangle motion and appearance Liu et al. (2024); Zhong et al.

(2024); Yu et al. (2024), we ensure temporal continuity by separating the prediction of long-range motion (keyframes) from short-range motion (interpolation). This approach allows us to model the video's temporal dynamics directly without requiring auxiliary losses Mukhopadhyay et al. (2024), perceptual models Li et al. (2024), or motion-specific frames Bigioi et al. (2024).

| | Method | CMMD ↓ | TOPIQ ↑ | VL ↑ | FVD ↓ | LipScore ↑ | Lipleak ↓ | Elo ↑ |
|---|---|---|---|---|---|---|---|---|
| Reconstruction | DiffDub Liu et al. (2024) | 0.403 | 0.44 | 37.12 | 429.07 | 0.34 | - | 1014 |
| | IP-LAP Zhong et al. (2023) | 0.091 | 0.49 | 37.77 | 282.02 | 0.36 | - | 1007 |
| | Diff2Lip Mukhopadhyay et al. (2024) | 0.225 | 0.48 | 35.84 | 555.08 | 0.49 | - | 886 |
| | TalkLip Wang et al. (2023a) | 0.230 | 0.39 | 29.07 | 608.92 | **0.58** | - | 920 |
| | LatentSync Li et al. (2024) | 0.319 | 0.41 | 45.23 | 343.90 | 0.52 | - | 1052 |
| | KeySync | **0.064** | **0.58** | **70.32** | **191.21** | 0.46 | - | **1120** |
| Cross-sync | DiffDub Liu et al. (2024) | 0.408 | 0.44 | 37.05 | 420.66 | 0.34 | 0.56 | 947 |
| | IP-LAP Zhong et al. (2023) | 0.093 | 0.49 | 35.32 | 294.66 | 0.17 | 0.57 | 1031 |
| | Diff2Lip Mukhopadhyay et al. (2024) | 0.231 | 0.48 | 33.97 | 601.68 | 0.16 | 0.42 | 878 |
| | TalkLip Wang et al. (2023a) | 0.201 | 0.42 | 24.80 | 704.93 | 0.30 | 0.90 | 911 |
| | LatentSync Li et al. (2024) | 0.325 | 0.41 | 45.95 | 361.57 | 0.14 | 0.64 | 1086 |
| | KeySync | **0.070** | **0.58** | **73.04** | **206.32** | **0.48** | **0.22** | **1145** |

Table 1: Quantitative comparison with other works on reconstruction and cross-synchronization performance. The best results are highlighted in **bold**, while the second-best results are underlined. All metrics are described in Section 4.2.

**Architecture.** Both stages use the Stable Video Diffusion (SVD) Blattmann et al. (2023) architecture. Crucially, unlike static image diffusion models, SVD employs a 3D U-Net that processes temporal sequences. The input to each stage consists of reference frames, the target audio, and the original video frames. The reference frames serve to either condition the interpolation or preserve identity. We use HuBERT Hsu et al. (2021) to extract audio embeddings, injected into the model's U-Net via cross-attention layers and timestep embeddings, enhancing video-audio alignment. Furthermore, we employ a modified classifier-free guidance strategy that decouples audio and identity conditions, which we found significantly boosts lip-synchronization accuracy (see Appendix I).

**Stage I: Keyframes.** This stage generates a sparse set of keyframes, $\{\hat{x}_{t_k}\}_{k=1}^T$, where each keyframe is spaced $S$ frames apart ($t_k = k \cdot S$). These keyframes serve as anchor points, ensuring that each one accurately reflects the phonetic content of the audio while preserving the subject's identity. In this stage, the reference input consists of an identity frame, randomly sampled from the source video and repeated $T$ times. To improve generalization, we augment these reference frames with noise Ho et al. (2022) and standard image augmentations.

**Stage II: Interpolation.** This stage interpolates between successive keyframes to achieve smooth, coherent motion. The reference frames takes two consecutive keyframes in the latent space, $\hat{z}_{t_i}$ and $\hat{z}_{t_{i+1}}$, and constructs the following input sequence to generate the intermediate frames:

$$s = \{z_{t_i}, \underbrace{z_m, \ldots, z_m}_{\text{repeat } S \text{ times}}, z_{t_{i+1}}\}, \tag{3}$$

where $z_m$ is a learnable embedding (optimized during training) that acts as a placeholder for the frames to be generated. This allows the model to "bridge" the motion between the fixed start and end points smoothly.

### 3.4 Losses

We adopt the loss formulation from Karras et al. 2022 applied to the video frames:

$$\mathcal{L}_{latent} = \mathbb{E}_{x,c,t,\sigma}\left[w_t \, \|F_\theta(z_t^m; c, \sigma_t) - z_t\|_2^2\right], \tag{4}$$

where $w_t$ is a weighting function, $F_\theta$ is the model, $\sigma_t$ is the noise level, and $c$ the conditioning inputs (audio and reference frames). We find that this loss alone is sufficient to achieve good lip synchronization and high-quality video generation. However, working solely in the compressed latent space can make it difficult for the model to retain fine semantic details Zhang et al. (2023b), which are critical for real-world lip synchronization tasks where preserving the nuances of the mouth region is essential. To address this, we introduce an additional $L_2$ loss in the RGB space. This requires decoding the latent output using the VAE decoder $\mathcal{V}$, resulting in:

$$\mathcal{L}_{rgb} = \mathbb{E}_{x,c,t,\sigma} \left[ w_t \left\| \mathcal{V}(F_\theta(z_t^m; c, \sigma_t)) - x_t \right\|_2^2 \right]. \tag{5}$$

The final combined loss is then:

$$\mathcal{L}_{total} = M \cdot \lambda(t)(\mathcal{L}_{latent}(\hat{z}, z) + \lambda_2 \mathcal{L}_{rgb}(\hat{x}, x)), \tag{6}$$

where $\lambda(t)$ is a weighting factor dependent on the diffusion timestep $t$, as defined in EDM Karras et al. (2022). Importantly, we ensure that only the generated region contributes to the loss computation by masking the region of interest. Note that while the losses are for reconstruction, they effectively enforce audio-visual alignment: since the ground-truth targets are synchronized, the model must utilize the audio condition to correctly predict the target mouth shape and minimize the reconstruction error.

### 3.5 Handling Occlusions

Occlusions are a critical yet often overlooked challenge in lip synchronization. Even advanced models can produce unnatural results if occlusions in the original video, such as a hand or microphone covering the mouth, are not properly accounted for. A common issue arises when an occlusion overlaps with the mouth region during masking, often causing the model to incorrectly generate the mouth over the occluding object, resulting in unnatural boundary artifacts.

To address this, we propose an inference-time solution to handle any occlusion without retraining. Explicitly training a model for occlusion handling is impractical due to the vast range of possible occlusions and their inherent misalignment with speech, making them hard for the model to learn. Instead, we introduce a preprocessing pipeline that first segments the occluding object using a state-of-the-art zero-shot video segmentation model Ravi et al. (2024), generating a mask $M_{obj}$ of the occlusion. We then refine the original mask M by excluding the occlusion:

$$M' = M \cap \neg M_{obj}, \tag{7}$$

where $\cap$ denotes intersection and $\neg$ denotes logical negation. Since our model supports free-form masks, as in RePaint Lugmayr et al. (2022), it can seamlessly reconstruct the mouth region while preserving the occluding object, ensuring visually coherence.

We purposefully employ a modular design using an off-the-shelf segmenter (SAM2 Ravi et al. (2024)) rather than training an end-to-end occlusion-aware model. During inference, the model is prompted by providing a point coordinates on the occluding object in a single frame, which SAM2 then propagates temporally across the sequence. This ensures our framework remains agnostic to segmentation improvements, as segmentation SOTA improves, KeySync improves without retraining. Our prompting strategy targets only foreground objects distinct from the facial manifold. Furthermore, the diffusion model's semantic priors prevent it from inpainting non-mouth textures into the mouth region even if the mask boundary is imperfect.

## 4 Experiments

### 4.1 Datasets

We train on a combination of HDTF Zhang et al. (2021), CelebV-HQ Zhu et al. (2022), and CelebV-Text Yu et al. (2023). To address artifacts in CelebV-HQ and CelebV-Text (e.g., low-quality, poor framing), we developed a data curation pipeline, which is detailed in Appendix A.

For evaluation, we focus on the cross-sync task, the primary use case for lip-sync models, where the input audio comes from a different video than the one being generated. We randomly select 100 test videos from

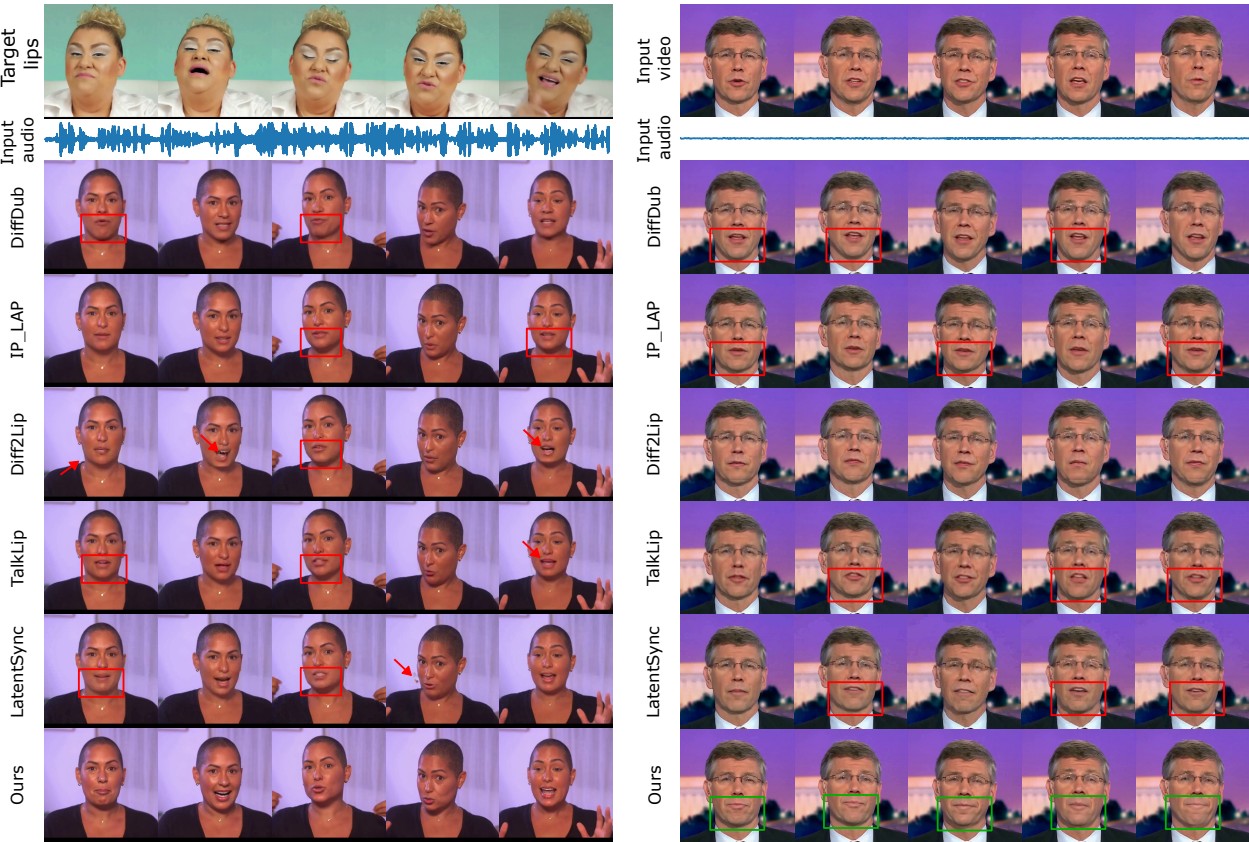

Figure 3: Qualitative comparison. "Target lips" (top row) shows the ground truth lip movements for the input audio.

Figure 4: Qualitative leakage comparison. We condition the models on silent audio and non-silent video (first row).

CelebV-Text, CelebV-HQ, and HDTF and swap their audio tracks. Additionally, to ensure consistency with prior works, we also report reconstruction results for the same 100 videos.

## 4.2 Evaluation Metrics

We evaluate our method using a set of no-reference metrics. For image quality, we measure the variance of Laplacian (VL) Pech-Pacheco et al. (2000) to assess blurriness, along with CMMD Jayasumana et al. (2024), an improved version of FID, and a facial-domain TOPIQ Chen et al. (2024a); Chen & Mo (2022). For video quality, we use FVD Unterthiner et al. (2019). For lip synchronization, we rely on LipScore Bigata et al. (2025), which correlates better with human perception than SyncNet Chung & Zisserman (2016). LipScore computes the cosine similarity between embeddings extracted by Auto-AVSR Ma et al. (2023), a state-of-the-art lipreading model. Specifically, these embeddings are derived from the mouth regions of both the generated frames and the ground-truth video (the source of the target audio). Notably, the underlying Auto-AVSR Ma et al. (2023) backbone resizes all inputs to $96 \times 96$, ensuring that the metric evaluates lip motion fidelity independently of the generation resolution. We also introduce LipLeak, detailed below, to quantify expression leakage. For completeness, SyncNet results are included in Appendix H.

**LipLeak** We introduce LipLeak to quantify expression leakage from a source video. We drive a model with both speech and silent audio; since silent audio provides a zero-signal ground truth, any resulting mouth motion is considered a leakage artifact. LipLeak is the ratio of the Mouth Aspect Ratio (MAR) Kannan

et al. (2023) standard deviation ($\sigma$) between the silent and speech-driven outputs:

$$\text{LipLeak} = \frac{\sigma(MAR_{silence})}{\sigma(MAR_{speech}) + \epsilon}, \qquad (8)$$

where $\epsilon$ ensures numerical stability. A low score is desirable, indicating expressive movement during speech and stability during silence. Conversely, a high score signals a problem, diagnosable by inspecting the components: a low $\sigma(MAR_{speech})$ suggests the model fails to generate expressive motion for speech, while a high $\sigma(MAR_{silence})$ points to instability or leakage, which manifests as unwanted mouth movement during silent periods. See Appendix E for further details.

### 4.3 User Study

While the metrics above offer an objective evaluation, they do not always align with human perception. To address this, we conduct a user study where participants compare randomly selected video pairs based on lip synchronization, temporal coherence, and visual quality. We then rank the performance of each model using the Elo rating system Elo (1978), and apply bootstrapping Chiang et al. (2024) for robustness. Further details are provided in Appendix G.

## 5 Results

This section presents a comprehensive evaluation of our model's performance against baselines, along with ablations to assess the impact of key components. Additional results are in Appendix H.

### 5.1 Comparison With Other Works

**Quantitative Analysis.** We evaluate our method alongside five competing approaches in Table 1. The evaluation is conducted in two settings: reconstruction, where videos are generated using the same audio as in the original video, and cross-sync, where the audio is taken from a different video. The latter is particularly relevant as it better reflects real-world applications such as automated dubbing, where the driving audio is typically not aligned with the input video.

As shown in Table 1, KeySync achieves superior visual quality and temporal consistency (VL, FVD) in both tasks. While most methods' lip-sync quality (LipScore) degrades in the more challenging cross-sync setting, our performance remains stable. Some baselines achieve a high LipScore in the reconstruction task, but this is an artifact of expression leakage, confirmed by their high LipLeak scores. For instance, LipLeak reveals that DiffDub's high cross-sync LipScore stems from random, unsynchronized mouth movements. Crucially, KeySync also obtains the highest Elo rating in our user study, confirming that our improvements in leakage reduction and visual quality translate directly to human preference (further analysis in Appendix E).

**Qualitative Analysis.** Figure 3 shows a qualitative cross-sync comparison. KeySync more accurately follows the lip movements corresponding to the input audio. While LatentSync and Diff2Lip also appear to align somewhat with the target lip movements, they fail to generate certain vocalizations correctly and exhibit visual artifacts (highlighted on the figure via red squares and arrows, respectively), limiting their practical usability. Additionally, most methods produce insufficient mouth movement. This can be attributed to expression leakage, where conflicting signals from the source video and new audio hinder the generation of a coherent mouth region.

**Leakage.** As discussed in Section 4.2, we compute LipLeak by generating a video using a silent audio input. Since the audio contains no speech, the mouth should exhibit minimal movement. However, in practice, we observe this is not always the case, as expressions from the input video can leak into the generated output. Figure 4 shows qualitative examples where all methods, except ours and Diff2Lip, exhibit several frames where the mouth is open (highlighted by red squares) due to expression leakage. While Diff2Lip manages to keep the mouth closed, it introduces significant blending artifacts, highlighting the model's struggle to suppress the original video's motion. In Figure 5, we visualize the standard deviation of

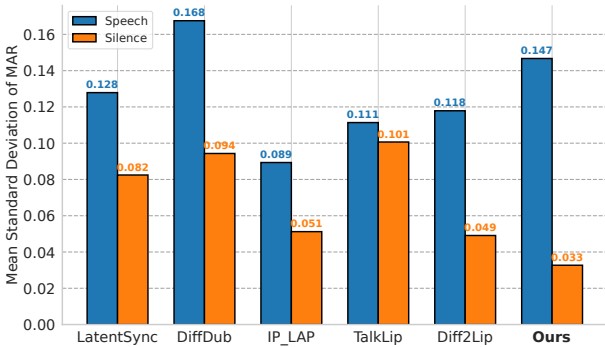

Figure 5: We show the mean standard deviation of MAR for silent and speech audios.

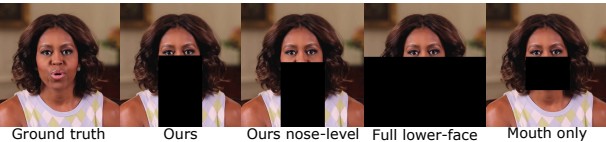

Figure 6: Examples of different masking techniques.

the Mouth Aspect Ratio (MAR) for both silent and speech audios. The results show that baselines either produce unwanted motion during silence (e.g., DiffDub, LatentSync), suppressed motion during speech (e.g., IP-LAP), or similar motion in both cases (e.g., TalkLip). In contrast, KeySync exhibits the desired behavior: high motion variability for speech and minimal motion for silence, confirming its robustness against leakage.

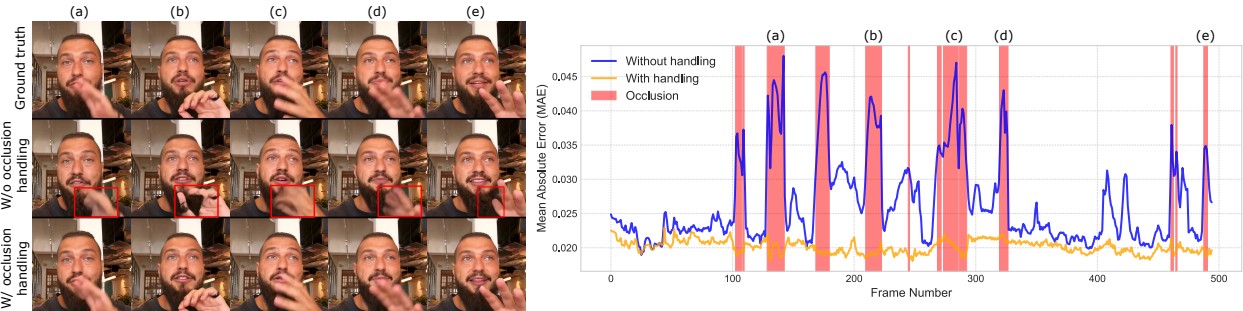

Figure 7: We present occlusion qualitative results on the left and quantitative results on the right.

**Occlusion Handling.** Figure 7 demonstrates our method's effectiveness. Without occlusion handling, significant artifacts appear around the hand (left), a finding confirmed by spikes in the mean absolute error plot (right). Our proposed method eliminates these artifacts by preserving the occluding object while maintaining correct lip synchronization, resulting in a much lower reconstruction error. We assess this technique on baseline models in Appendix F.

## 5.2 Ablation Studies

| Audio backbone | FVD ↓ | LipScore ↑ | Lipleak ↓ |
|---|---|---|---|
| Whisper Radford et al. (2023) | 207.41 | 0.47 | 0.25 |
| Wav2vec2 Baevski et al. (2020) | **201.13** | 0.45 | 0.26 |
| WavLM Chen et al. (2022) | 218.08 | **0.48** | 0.23 |
| HuBERT Hsu et al. (2021) | 206.32 | **0.48** | **0.22** |

Table 2: Audio encoder ablation in the cross-sync setting.

**Audio Encoder.** We also investigate the impact of different audio encoders on the generated videos, as shown in Table 2. We see that Wav2vec2 Baevski et al. (2020) produces marginally higher video quality, as indicated by its lower FVD score. However, this comes at the expense of lip synchronization, as reflected in

its lower LipScore. With WavLM Chen et al. (2022), we achieve a LipScore comparable to HuBERT Hsu et al. (2021), but at the cost of worse video quality. In contrast, HuBERT maintains a strong LipScore and achieves the lowest LipLeak, indicating effective mitigation of expression leakage. Therefore, we select HuBERT as our default audio encoder.

| Keyframe generator | CMMD ↓ | FVD ↓ | LipScore ↑ |
|---|---|---|---|
| Only (one-stage) | 0.085 | 395.45 | 0.32 |
| Image-based | 0.142 | 618.27 | 0.39 |
| Sequence-based | **0.070** | **206.32** | **0.48** |

Table 3: Keyframe generator ablation in cross-sync setting.

| Mask | CMMD ↓ | FVD ↓ | LipScore ↑ | Lipleak ↓ |
|---|---|---|---|---|
| Mouth-only | 0.077 | 200.71 | 0.23 | 0.52 |
| Full lower-face | 0.743 | 219.96 | 0.35 | 0.38 |
| Nose-level | 0.071 | **199.39** | 0.34 | 0.48 |
| Ours | **0.070** | 206.32 | **0.48** | **0.22** |

Table 4: Mask ablation in cross-sync setting.

**Keyframe generator.** We evaluate our keyframe/interpolation approach against two alternative designs. The first is a one-stage model that generates frames sequentially without an interpolation model; longer videos are formed by concatenating the generated clips with a one-frame overlap. The second retains the two-stage design but generates keyframes individually with an image-based model, skipping the temporal modelling of our approach. We find that the one-stage model, while achieving reasonable visual quality (CMMD), suffers a sharp decline in FVD and LipScore, underscoring the value of our interpolation strategy for generating smooth, well-synchronized motion. Likewise, generating keyframes individually without temporal context degrades long-range coherence, leading to a significant drop across all metrics.

**Mask.** Finally, we investigate the impact of different masking techniques (illustrated in Figure 6) in Table 4. A mouth-only mask improves video quality by minimizing obstruction but causes severe leakage (low LipScore, high LipLeak), as the model tracks the mask's motion rather than syncing with the audio. Conversely, masking the entire lower face effectively reduces leakage but severely harms image and video quality, as the model must reconstruct unrelated background elements. Our proposed box-style mask offers a balanced trade-off, achieving the best overall performance. We found that extending the mask to cover the cheeks is crucial for maximizing LipScore, as this region conveys important cues about mouth movements that can otherwise cause leakage. We further discuss the implications of the baselines' masking strategies on leakage in Appendix D.

## 6 Conclusion

In this paper, we propose KeySync, a state-of-the-art lip synchronization approach based on a two-stage video diffusion model. We show that, unlike other methods, KeySync generates high-resolution videos which are temporally coherent and closely aligned with the driving audios. Furthermore, by applying a new masking strategy, we show that our model successfully minimizes expression leakage from the input video, while also being robust to facial occlusions that may occur in the wild. We hope that these improvements will enable the use of lip synchronization models in applications such as automated dubbing, which can help eliminate language barriers at scale.

**Compute and Institutional Note.** All experiments reported in this paper, including model training, evaluation, and ablation studies, were conducted using computational resources at Imperial College London.

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

# A  Datasets

## A.1  Overlap with SVD dataset

While the SVD backbone was pre-trained on LVD-10M Blattmann et al. (2023) (general video), our fine-tuning datasets are strictly domain-specific; given the specific curation of our talking-head data, we estimate negligible overlap with the pre-training corpus.

## A.2  Curation and Preprocessing

When working with in-the-wild datasets such as CelebV-HQ Zhu et al. (2022) and CelebV-Text Yu et al. (2023), we observed that a significant portion of the data is of suboptimal quality. Common issues include visible hands, camera movement, editing artifacts, and occlusions. Additionally, some samples exhibit lower resolution than advertised. Examples of these issues are illustrated in Figure 8. During training, we found that such videos negatively impacted model performance because their visual content correlates poorly with the corresponding audio. To address these challenges, we developed a data curation pipeline comprising the following steps:

- Extract videos at 25 FPS and single-channel audio at 16 kHz.

- Discard low-quality videos based on HyperIQA Su et al. (2020) scores below 0.4. Each video's score is computed as the average of nine evaluations: selecting the first, middle, and last frames, each evaluated on three random crops.

- Detect and segment scenes using PySceneDetect[1].

- Remove clips without active speakers using Light-ASD Liao et al. (2023) indicated by the score below 0.75.

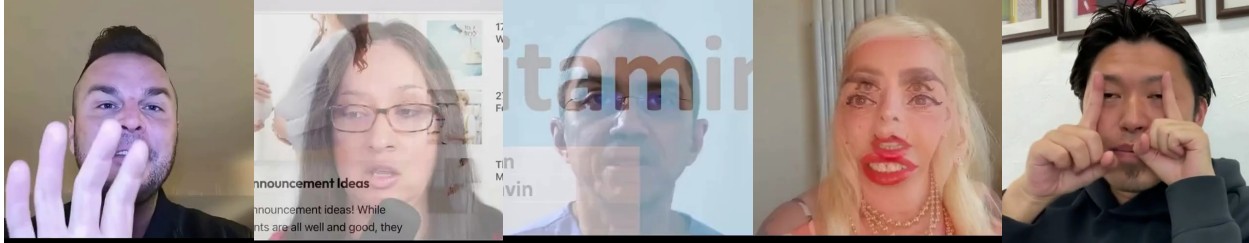

Figure 8: Examples of problematic videos in CelebV-HQ and CelebV-Text.

## A.3  Data Statistics

Table 5 describes the training/evaluation data used in this paper, specifying the number of speakers, videos, average video duration, and total duration for each dataset. Additionally, to illustrate the impact of our data curation pipeline, we present Table 6, which details the statistics of the datasets before curation. Overall, we discard roughly 75 % of the original videos. Please note that CelebV-HQ and CelebV-Text videos were split into shorter chunks during pre-processing, hence the higher video count in Table 5.

# B  Inference pipeline

The complete inference procedure is detailed in Algorithm 1. We process the input video in latent space, first generating robust masks that integrate facial landmarks with SAM2-based occlusion segmentation. The generation then proceeds in two stages to balance temporal stability with motion fidelity: the Keyframe

---

[1]https://github.com/Breakthrough/PySceneDetect

| Dataset | # Speakers | # Videos | Duration | |
|---|---|---|---|---|
| | | | Avg. (sec.) | Total (hrs.) |
| HDTF | 264 | 318 | 139.08 | 12 |
| CelebV-HQ | 3,668 | 12,000 | 4.00 | 13 |
| CelebV-Text | 9,109 | 75,307 | 6.38 | 130 |

Table 5: Data statistics after curation and pre-processing.

| Dataset | # Videos | Duration | |
|---|---|---|---|
| | | Avg. (sec.) | Total (hrs.) |
| HDTF | 318 | 139.08 | 12 |
| CelebV-HQ | 35,666 | 6.86 | 68 |
| CelebV-Text | 70,000 | 14.35 | 279 |

Table 6: Data statistics before curation.

Stage generates sparse, audio-aligned anchor frames (every $S = 12$ frames) to establish long-term consistency, followed by the Interpolation Stage, which synthesizes the intermediate frames conditioned on the generated anchors and the corresponding audio interval. Finally, the overlapping segments are merged and decoded to produce the high-resolution synchronized output.

---

**Algorithm 1** KeySync Inference Pipeline

---

**Require:** Source Video $V_{src}$, Driving Audio $A_{drv}$, Occlusion Prompt $P_{occ}$ (optional)
**Ensure:** Synchronized Video $V_{out}$
   **Models:** Keyframe Model $\mathcal{M}_{KF}$, Interpolation Model $\mathcal{M}_{INT}$, SAM2 $\mathcal{S}$, VAE $\mathcal{E}/\mathcal{D}$
                                                                          ▷ **1. Preprocessing & Mask Generation**
1: $Z_{src} \leftarrow \mathcal{E}(V_{src})$                                       ▷ Encode video to latent space
2: $E_{audio} \leftarrow \text{AudioEncoder}(A_{drv})$                        ▷ Extract Wav2Vec2/WavLM features
3: $M_{face} \leftarrow \text{Landmarks}(V_{src})$                              ▷ Generate lower-face masks
4: **if** $P_{occ}$ is provided **then**                                   ▷ Occlusion Handling (SAM2)
5:     $M_{occ} \leftarrow \mathcal{S}(V_{src}, P_{occ})$                     ▷ Segment occlusion (e.g., hand)
6:     $M \leftarrow M_{face} \odot (1 - M_{occ})$                          ▷ Exclude occlusion from mask
7: **else**
8:     $M \leftarrow M_{face}$

                                                          ▷ **2. Stage I: Keyframe Generation (Sparse Anchors)**
9: $Z_{KF} \leftarrow []$
10: **for** $t \leftarrow 0$ **to** length$(Z_{src})$ **step** $S$ **do**
11:     $z_{masked} \leftarrow Z_{src}[t] \odot (1 - M[t])$                    ▷ Mask the mouth region at step $t$
12:     $c_{aud} \leftarrow E_{audio}[t]$                                   ▷ Get audio embedding for frame $t$
13:     $\hat{z}_t \leftarrow \mathcal{M}_{KF}(z_{masked}, c_{aud})$            ▷ Generate coherent anchor frame
14:     Append $\hat{z}_t$ to $Z_{KF}$

                                                          ▷ **3. Stage II: Interpolation (High-Frequency Motion)**
15: $Z_{out} \leftarrow []$
16: **for** $i \leftarrow 0$ **to** length$(Z_{KF}) - 1$ **do**
17:     $z_{start}, z_{end} \leftarrow Z_{KF}[i], Z_{KF}[i+1]$                 ▷ Set start/end anchors
18:     $c_{seq} \leftarrow E_{audio}[t_{start} : t_{end}]$                   ▷ Get interval audio sequence
19:     $\hat{Z}_{seg} \leftarrow \mathcal{M}_{INT}(z_{start}, z_{end}, c_{seq})$   ▷ Fill intermediate frames
20:     $Z_{out} \leftarrow \text{Merge}(\hat{Z}_{seg}, Z_{out})$
21: **return** $\mathcal{D}(Z_{out})$                                        ▷ Decode to pixel space

---

# C  Implementation Details

**Code**  The code and model weights will be released upon acceptance.

**Hyperparameters & Training Configuration**  We summarize all the hyperparameters of our pipeline in Table 7. The weights of the U-Net and VAE are initialized from SVD Blattmann et al. (2023). The interpolation model undergoes more training steps because its task differs more significantly from the original task of SVD. The final hyperparameters were selected through extensive experimentation to find the optimal trade-off between lip-synchronization accuracy (LipScore), visual quality (FVD), and expression leakage (LipLeak).

To optimize memory efficiency, we apply $\mathcal{L}_{rgb}$ to a randomly selected frame from the sequence, which we found to be sufficient for maintaining perceptual quality.

| Hyperparameter | Final Value | Range Tested |
|---|---|---|
| Keyframe seq. length ($T$) | 14 | Fixed |
| Keyframe spacing ($S$) | 12 | Fixed |
| Interpolation seq. length ($S$) | 12 | Fixed |
| Keyframe training steps | 60,000 | N/A |
| LipLeak $\epsilon$ | $10^{-5}$ | Fixed |
| Interpolation training steps | 120,000 | N/A |
| Training batch size | 32 | {16, 32} |
| Optimizer | AdamW | Fixed |
| Learning rate | $10^{-5}$ | {$10^{-4}$, $10^{-5}$} |
| Warmup steps | 1,000 | {1,000, 2,000} |
| Inference steps | 10 | Fixed |
| GPU used | A100 | N/A |
| Video frame rate | 25 | Fixed |
| Audio sample rate | 16,000 | Fixed |
| Resolution | $512 \times 512$ | Fixed |
| Pixel loss weighting ($\lambda_2$) | 1 | {0, 0.5, 1.0} |
| Audio cond. drop rate | 20% | {10%, 20%, 30%} |
| Identity cond. drop rate | 10% | {5%, 10%, 20%} |

Table 7: Final model hyperparameters and the ranges tested during development. "Fixed" denotes values set by the model architecture or data standards, while "N/A" denotes values not typically tuned.

**Practical Deployment**  A limitation of our model is its inference speed, which is not yet real-time. Nevertheless, our two-stage approach is faster than other diffusion-based methods (e.g., DiffDub, Diff2Lip, LatentSync) and competitive with some GANs, as shown in Table 8. This advantage stems from our framework's support for batched inference, a feature absent in autoregressive models.

| *Diffusion-based Methods* | | | | |
|---|---|---|---|---|
| **Model** | VideoReTalking | DiffDub | Diff2Lip | LatentSync | KeySync |
| **FPS** | 0.17 | 0.69 | 1.56 | 2.50 | **3.84** |
| *GAN-based Methods* | | | | |
| **Model** | IP-LAP | Wav2Lip | TalkLip | | |
| **FPS** | 4.31 | 16.66 | 92.00 | | |

Table 8: Inference speed comparison in Frames Per Second (FPS). Higher is better.

Future work could focus on acceleration by adapting techniques from recent literature. For example, Consistency Models Song et al. (2023) can enable single-step generation by learning to map any point on a

diffusion trajectory back to the origin. Other promising approaches, such as adversarial distillation Sauer et al. (2024), can also reduce a trained diffusion model to a single-step generator while maintaining high output quality.

We provide a breakdown of the computational cost for each stage of the pipeline in Table 9. The U-Net sampling dominates the inference time (63.0%), with the majority spent in the Interpolation Sampling stage (56.4%). This is expected, as this stage is responsible for generating the high-frequency motion details for the majority of the video frames. The segmentation module (SAM2) accounts for 18.6% of the total time; since this step is optional, it can be omitted for non-occluded videos to boost throughput, or replaced with a lighter-weight segmentation model in future iterations. Preprocessing (VAE encoding and landmark extraction) accounts for roughly 16% of the latency; in our training pipeline, we precompute these features to accelerate experimentation.

| Stage | Percentage (%) |
|---|---|
| VAE Encoding | 5.8 |
| Landmarks Extraction | 10.4 |
| Audio Preprocessing | 0.1 |
| – HuBERT Embeddings | 0.0 |
| – WavLM Embeddings | 0.0 |
| Segmentation (SAM2) | 18.6 |
| UNet Sampling | 63.0 |
| – Keyframe Sampling | 6.6 |
| – Interpolation Sampling | 56.4 |

Table 9: Timing Summary Breakdown

# D Masking

## D.1 Mask Definition

To create the mask defined in Section 3.2, we first compute 68 facial landmarks in 2D Bulat & Tzimiropoulos (2017) and then follow the procedure in Algorithm 2.

---

**Algorithm 2** Create mask from landmarks

---

**Require:** $L \in \mathbb{R}^{T \times K \times 2}$      ▷ landmarks for $T$ frames
  1: $(H, W)$      ▷ image height and width
  2: $n$ (nose index, default 28)
**Ensure:** $M \in \{0, 1\}^{T \times H \times W}$      ▷ binary masks
  3: $M \leftarrow \mathbf{0}_{T \times H \times W}$      ▷ initialise masks
  4: **for** $t \leftarrow 0$ **to** $T - 1$ **do**
  5:      $P \leftarrow L_t$      ▷ landmarks of frame $t$
  6:      $h_c \leftarrow P_{n,y}$      ▷ $y$-coord. of the nose
  7:      $l \leftarrow \arg\min_k P_{k,x}$      ▷ left-most landmark index
  8:      $r \leftarrow \arg\max_k P_{k,x}$      ▷ right-most landmark index
  9:      $p_1 \leftarrow (P_{l,x}, h_c)$
10:      $p_2 \leftarrow (P_{l,x}, H)$
11:      $p_3 \leftarrow (P_{r,x}, H)$
12:      $p_4 \leftarrow (P_{r,x}, h_c)$
13:      FILLPOLYGON($M_t$, $[p_1, p_2, p_3, p_4]$, 1)
14: **return** $M$

---

## D.2 Alternative Masking

In Figure 9, we illustrate the different masking strategies for the methods analysed in Section 5. We observe that while the masks of IP-LAP and Diff2Lip are closest to our own, their performance is affected by a tight facial crop applied before masking. This crop, which typically extends to the jawline, can leak the state of the mouth and excludes other speech-related areas such as the throat, which is reflected in their leakage scores in Table 1.

TalkLip masks the lower part of the image but fails to cover the cheek region, which contains important cues about the mouth's state, resulting in a high LipLeak score. The mask shape used by DiffDub is similar to ours, but because it does not extend to the bottom of the frame, the model can infer the mouth shape from the mask's position relative to the chin. Similarly, LatentSync uses a fixed mask and preprocesses the video so the mouth is always contained within it; however, this allows the model to infer mouth movements based on the position of the head rather than the audio content.

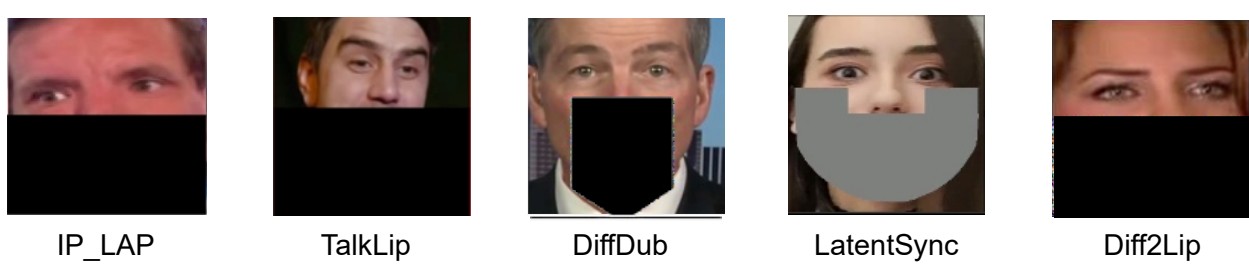

| IP_LAP | TalkLip | DiffDub | LatentSync | Diff2Lip |

Figure 9: Illustration of the masking strategy of baseline methods

# E Leakage metric

## E.1 MAR Calculation

We introduce LipLeak as part of our evaluation pipeline for measuring expression leakage. The first step in computing LipLeak is to calculate the Mouth Aspect Ratio (MAR) from facial landmarks, as illustrated in Figure 10. This ratio quantifies the vertical openness of the mouth relative to its width, increasing as the mouth opens wider. Because LipLeak is based on a ratio, it is a scale-invariant measure, allowing for consistent evaluation across different video resolutions and face sizes.

## E.2 Alternative Metric

While LipLeak is a reliable metric, it requires running the model twice (once with speech audio and once with silent audio). To create a simpler metric, we propose $\text{LipLeak}_{lite}$, which only requires a single run with silent audio. $\text{LipLeak}_{lite}$ measures the proportion of time the mouth is open when the audio is silent. We found that models opening their mouths during silent periods appear unnatural to users, making this a critical failure mode for real-world scenarios. To determine whether the mouth is open, we apply a threshold to the MAR; based on visual inspection, we selected a threshold of 0.25, as any MAR below this value consistently represents a closed mouth.

To validate $\text{LipLeak}_{lite}$, we first assessed its sensitivity to this threshold. As shown in Figure 11, $\text{LipLeak}_{lite}$ decreases smoothly and predictably as the threshold increases. This stable behaviour is essential for a reliable metric, as it prevents erratic jumps that could compromise quantitative evaluations. Finally, to ensure $\text{LipLeak}_{lite}$ effectively captures the same underlying issue as LipLeak, we computed the correlation between the two metrics in Figure 12. We observe a significant ($p < 0.05$) strong correlation between $\text{LipLeak}_{lite}$ and LipLeak, confirming that $\text{LipLeak}_{lite}$ is an efficient and reliable proxy for quantifying model leakage.

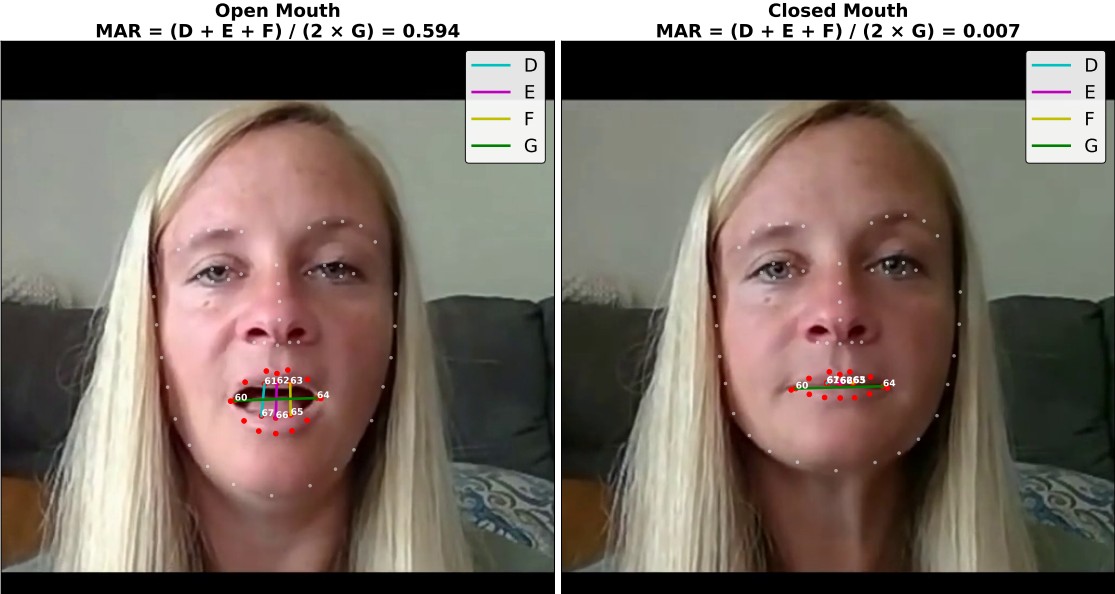

Figure 10: Mouth Aspect Ratio measurement example.

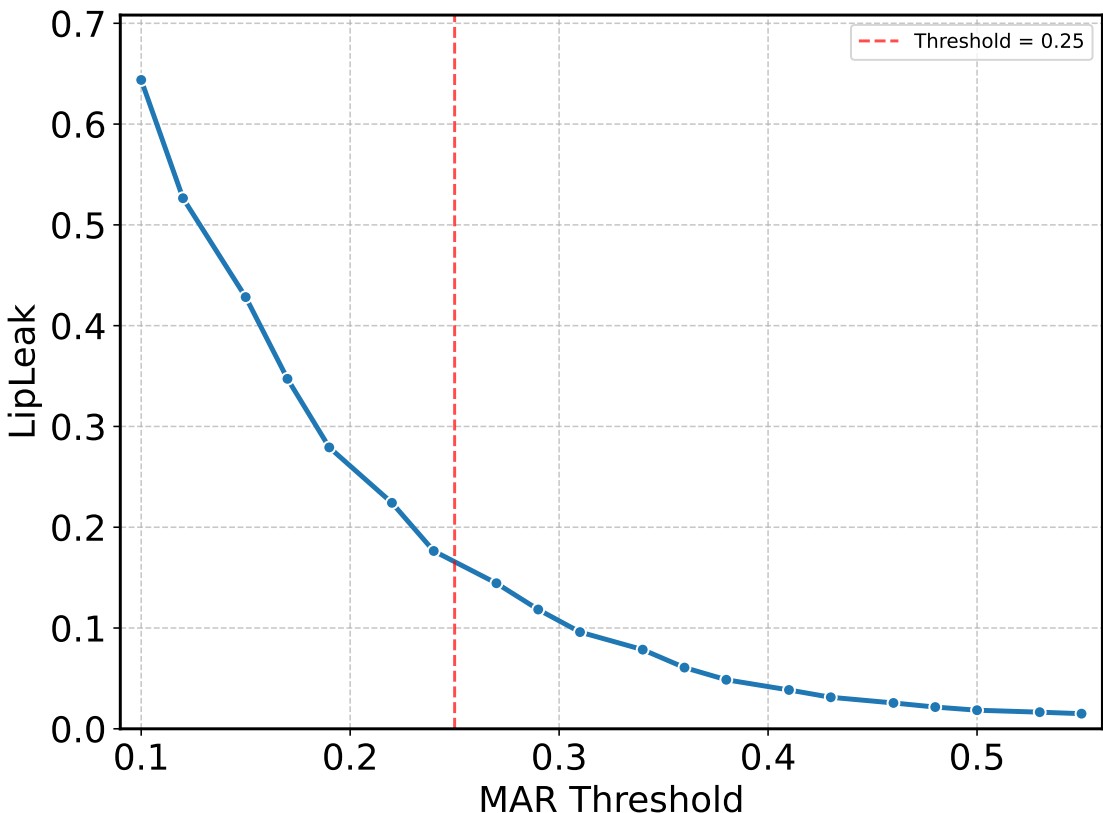

Figure 11: LipLeak as a function of the MAR threshold.

We note that a model could trivially achieve a low LipLeak score by producing minimal mouth motion for all inputs; however, such behavior would be heavily penalized by our primary sync metric, LipScore, which measures the positive correlation between audio and visual speech cues.

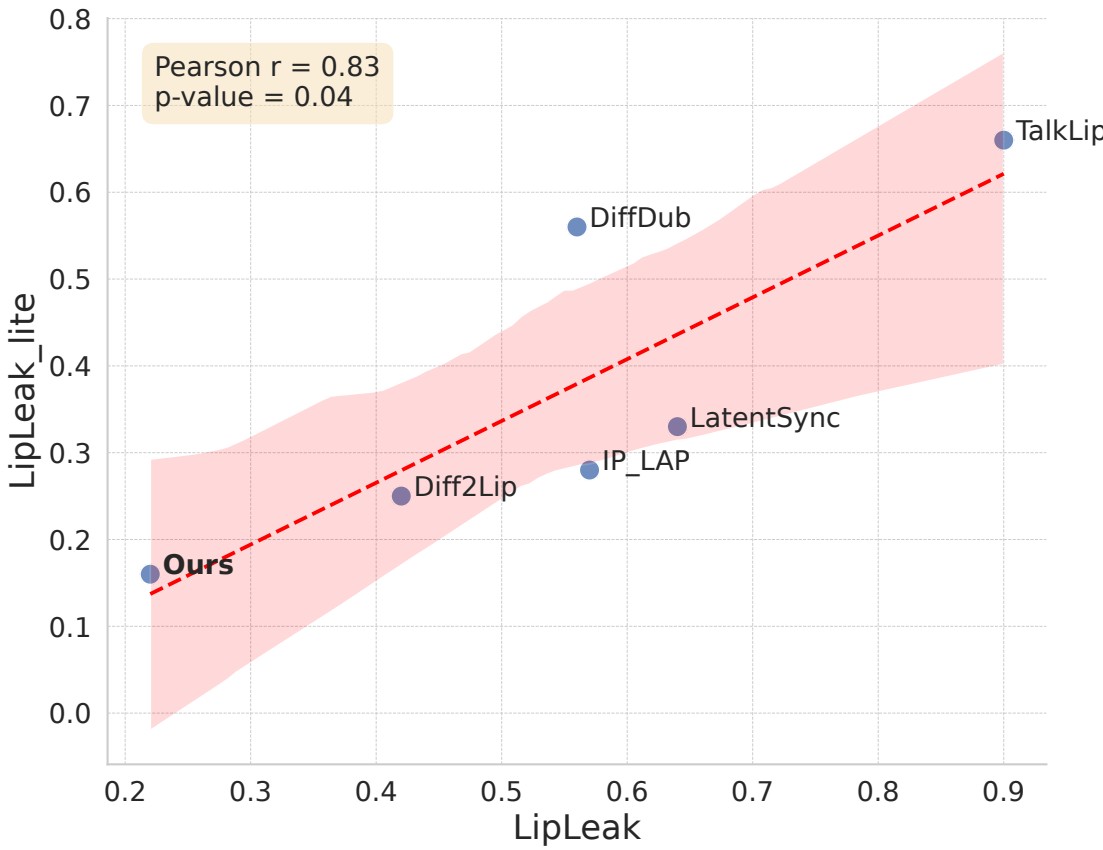

Figure 12: Correlation between Mouth Stability Metrics.

### E.3 Correlation with Human Perception

To validate the perceptual relevance of our novel metric, we analyse the relationship between LipLeak scores and human judgment. Figure 13 illustrates a strong negative correlation ($r = -0.79$) between LipLeak values and human ratings for "Overall Coherence". This result confirms that higher leakage (mouth movement during silence) is consistently perceived as a negative artefact by users. We note that *Diff2Lip* appears as an outlier in this trend. Qualitative inspection reveals that this method suffers from significant blending artifacts around the mouth region, which heavily penalize its human rating regardless of its leakage behaviour. This suggests that while minimizing leakage is a prerequisite for perceived coherence, it is not the sole factor; high visual fidelity and seamless blending are equally necessary for achieving state-of-the-art user satisfaction.

## F    Occlusion Handling

### F.1    Dependency on external segmentation model

Handling occlusions via an external, state-of-the-art segmentation model is a deliberate design choice that provides significant flexibility. This modular approach allows us to benefit from rapid advancements in video segmentation without architectural changes or retraining. Any improvement in segmentation technology can be directly integrated, immediately boosting the system's robustness. The trade-off is a dependency on this upstream component, as segmentation failures can propagate into the final result.

While complete segmentation failures are rare with SAM2, they constitute a known limitation of our pipeline. We categorize these failures into two types:

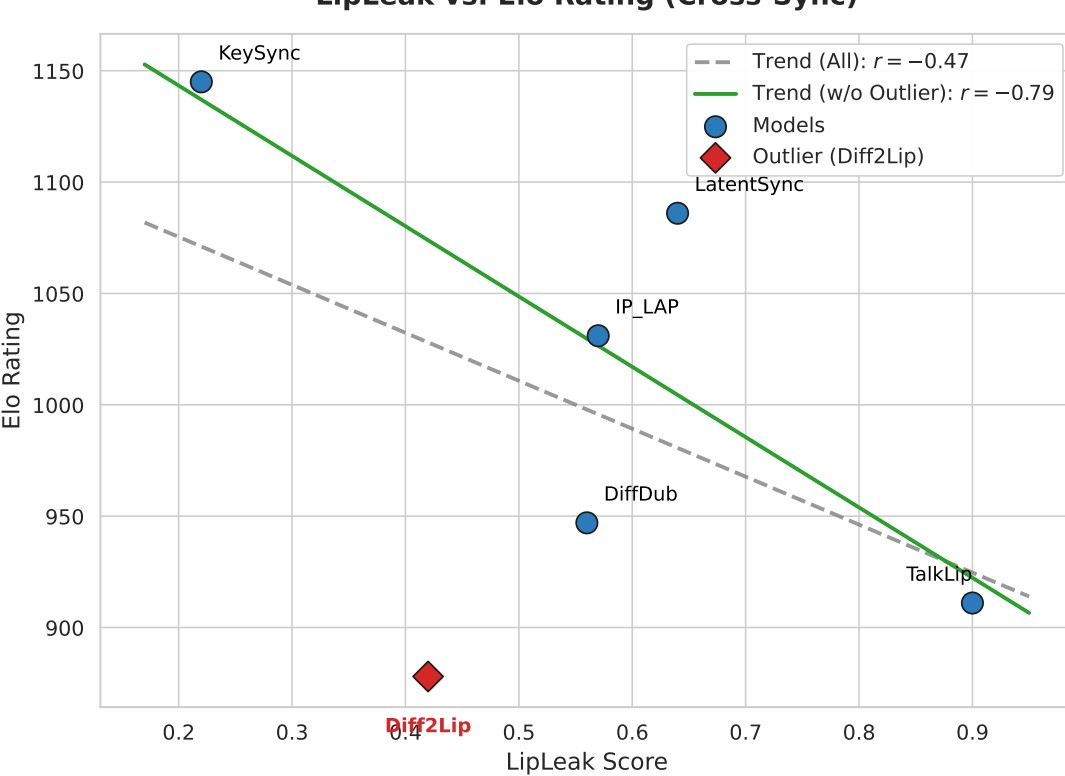

Figure 13: Correlation between LipLeak and human perception. The plot reveals a strong negative correlation ($r = -0.79$).

1. **Boundary Errors (Partial Failure):** For fast-moving objects or heavy motion blur, the segmentation mask may miss fine edge details. In these cases, we observe that the SVD backbone is generally robust enough to leverage video priors and fill small gaps plausibly, though minor high-frequency artifacts may persist at the boundaries (see Figure 14 top row).

2. **Total Segmentation Failure (False Negative):** If the segmentation model fails to detect the occluding object entirely, the occlusion mask $M_{obj}$ remains empty. Consequently, the final inpainting mask $M$ includes the object as part of the generation region. As the diffusion backbone is conditioned to generate a talking mouth, it will erroneously Hallucinate lip textures *over* the obstacle. We visualize this failure mode in Figure 14 (bottom row) by simulating a missed detection on a microphone occlusion.

## F.2   Application to different methods

Figure 15 illustrates the application of our occlusion handling technique to several existing methods:

- **DiffDub Liu et al. (2024) and Diff2Lip Mukhopadhyay et al. (2024):** Our approach works out of the box, seamlessly handling occlusions without requiring modifications.

- **LatentSync Li et al. (2024):** Since this method employs a fixed mask, the model has never been exposed to variations in masking. As a result, it struggles to adapt to the new mask patterns introduced by our occlusion-handling technique, highlighting a key drawback of using a rigid masking approach.

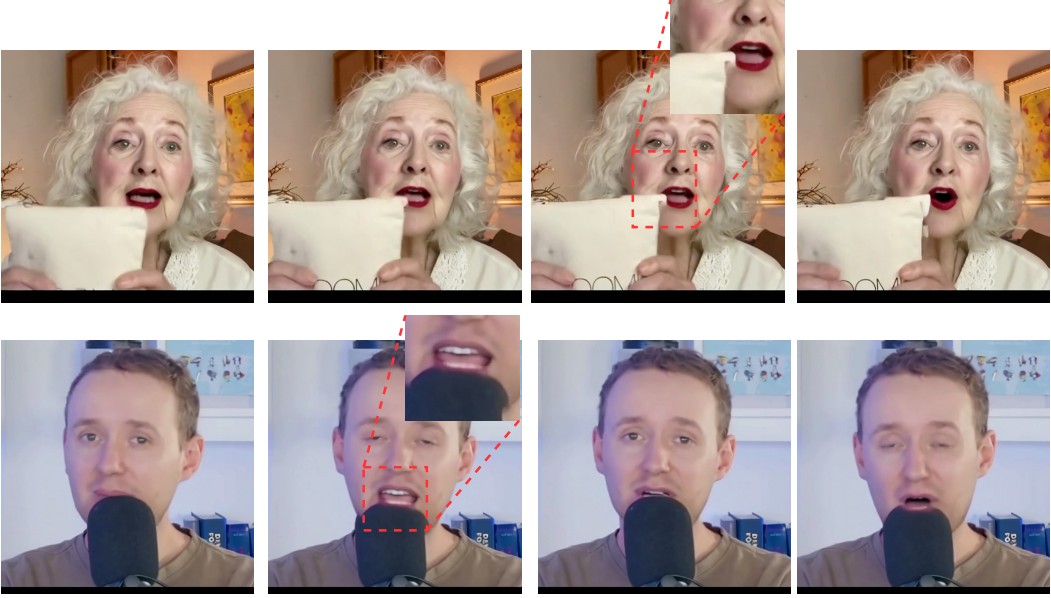

Figure 14: Occlusion Failure Case Analysis.

- **IP-LAP Zhong et al. (2023):** This model generates the mouth region separately through an audio-to-landmark module. Consequently, the occlusion mask has no direct effect, and the mouth is generated on top of the occlusion.

- **TalkLip Wang et al. (2023a):** At first glance, TalkLip appears to function without occlusion handling. However, it achieves this by concatenating frames from the original video to generate new frames. This shortcut enables occlusion handling but comes at the cost of significant expression leakage, as evidenced by its very high LipLeak score in Table 1.

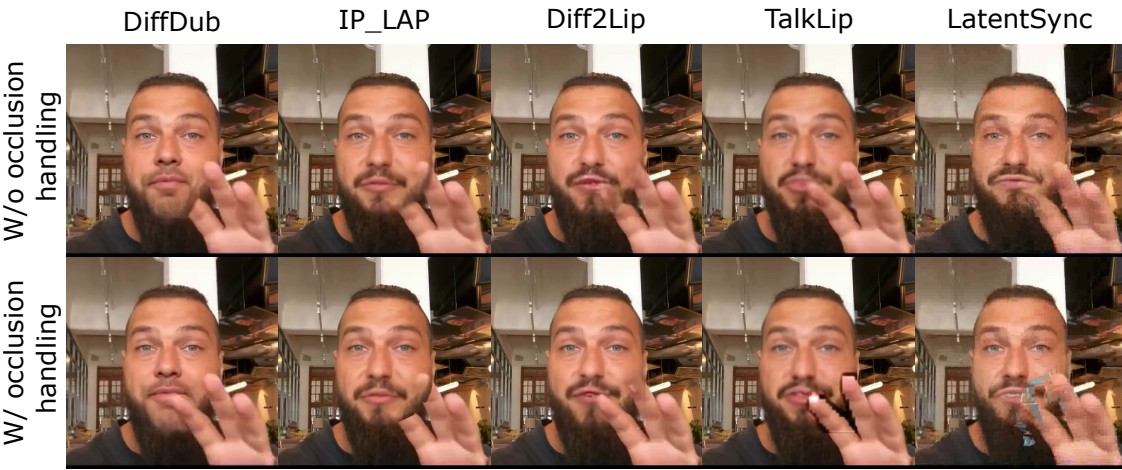

Figure 15: Effectiveness of Occlusion Handling Across Different Methods.

# G    User Study Results

To ensure that the objective metrics presented in Table 1 align with human perception, we conduct a user study to evaluate model performance in terms of lip synchronization, overall coherence, and image quality.

Participants are presented with pairs of videos and asked to select the one they preferred based on these criteria. The video pairs are randomly sampled from the pool of models listed in Table 1 to ensure a fair and unbiased comparison. A total of $1,000$ pairwise comparisons were collected, providing a robust dataset for evaluating human preferences. Figure 16 shows a screenshot of the user study interface, illustrating the evaluation setup.

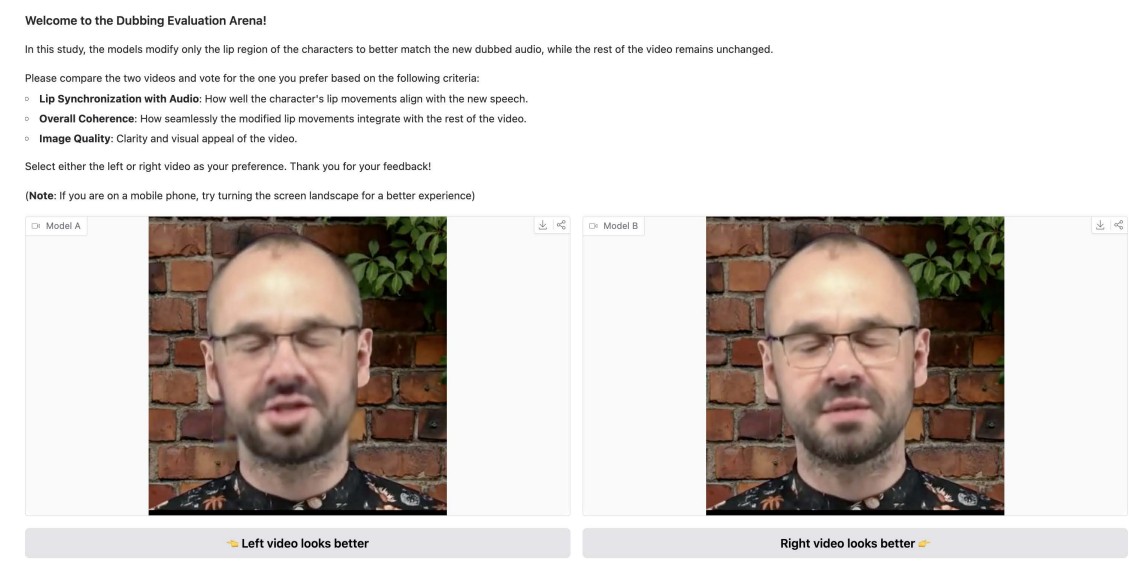

Figure 16: User study interface. Participants were shown side-by-side videos and asked to select the preferred one based on lip synchronization, coherence, and quality.

**Elo Ratings**  To assess the relative performance of different models in our evaluation framework, we employ the Elo rating system Elo (1978), a widely used method for ranking competitors based on pairwise comparisons. The Elo rating system assigns scores to models based on their performance in direct comparisons, updating their ratings dynamically as more results are collected.

We evaluate Elo ratings in two distinct settings:

- **Reconstruction setting (Figure 17):** In this scenario, we compare videos are generated using the same audio as in the original video.

- **Cross-Synchronization Setting (Figure 18):** In this scenario, we compare videos generated using a different audio from the original video.

In both cases, our model consistently outperforms competing methods, achieving higher Elo ratings. This demonstrates its superior ability to generate high-quality, accurately synchronised lip movements, both in the reconstruction and cross-synchronization tasks.

**Elo Rating Distributions**  To better understand the distribution and variance of model rankings, we analyse the overall Elo ratings across all evaluated models. Figure 19 presents a histogram of Elo scores, illustrating how models are ranked relative to each other. A well-separated distribution suggests clear performance differences between models, whereas overlapping scores indicate models with similar performance levels. Our model achieves the highest Elo ratings, forming a well-defined peak that highlights its superior performance. In contrast, baseline models display varying degrees of separation, with some exhibiting significant overlap, suggesting closer competition and comparable performance in certain cases.

**Win Rates**  Beyond Elo ratings, we compute win rates to assess how often each model outperforms others in pairwise comparisons. The win rate matrix in Figure 20 provides a detailed overview of direct matchups,

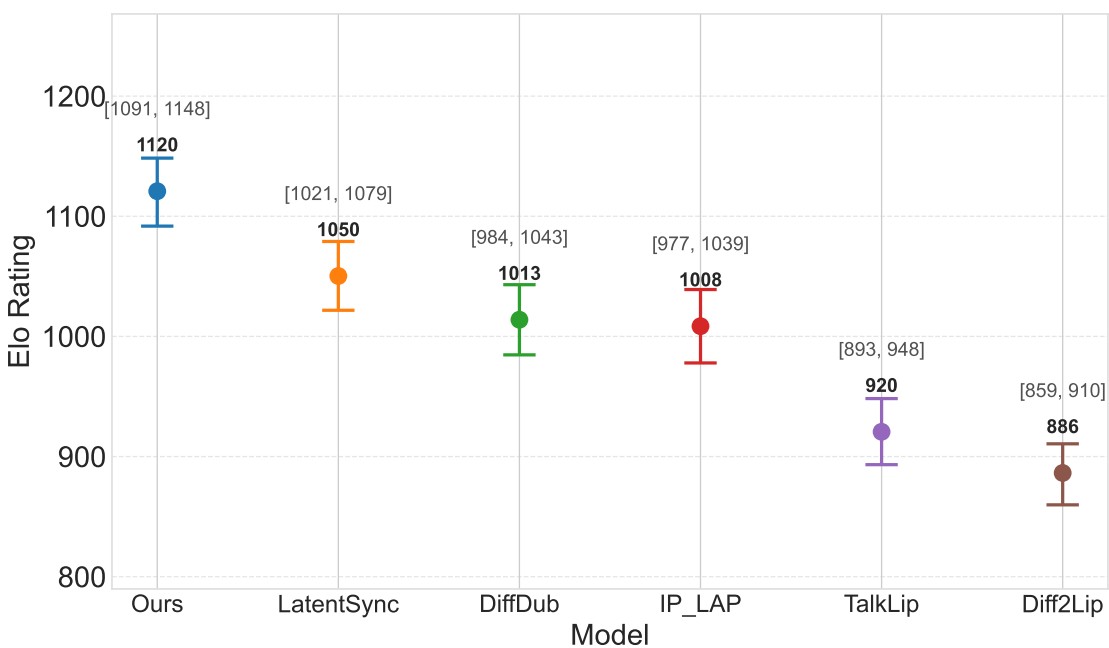

Figure 17: Elo ratings in the reconstruction setting. Higher ratings indicate better performance in generating videos with original audio as input.

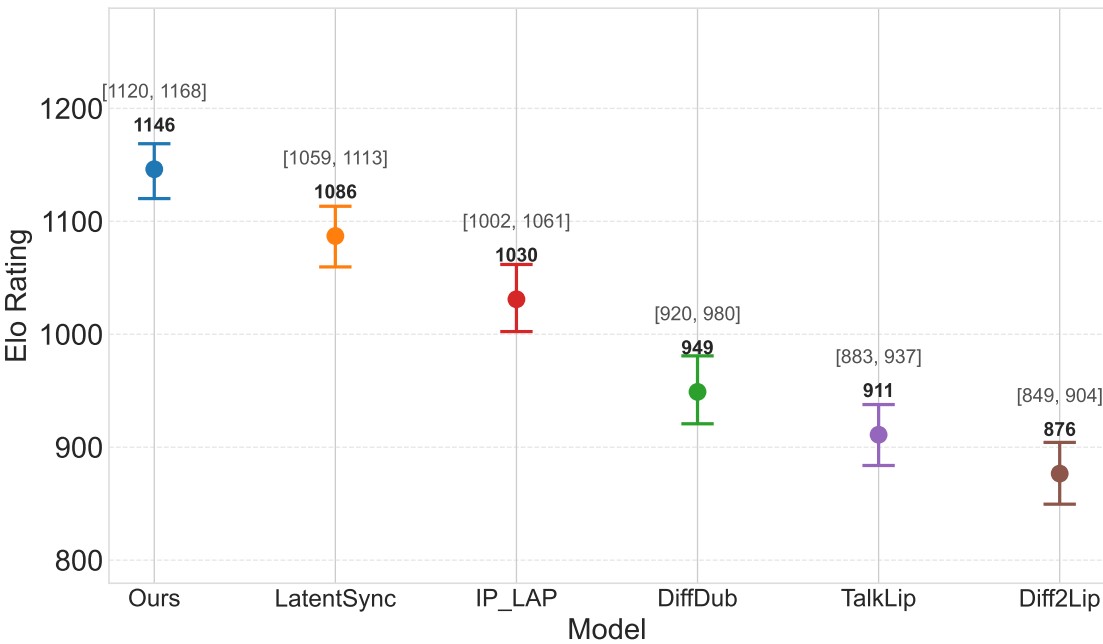

Figure 18: Elo ratings in the cross-sync setting. Higher ratings indicate better performance in generating videos with different audio from input.

where each cell represents the percentage of times one model wins against another. This analysis helps identify dominant models and potential inconsistencies in ranking. Our model consistently outperforms competing approaches, achieving a minimum win rate of 69 % and a maximum of 94 %. These results indicate a strong and reliable performance advantage over alternative methods.

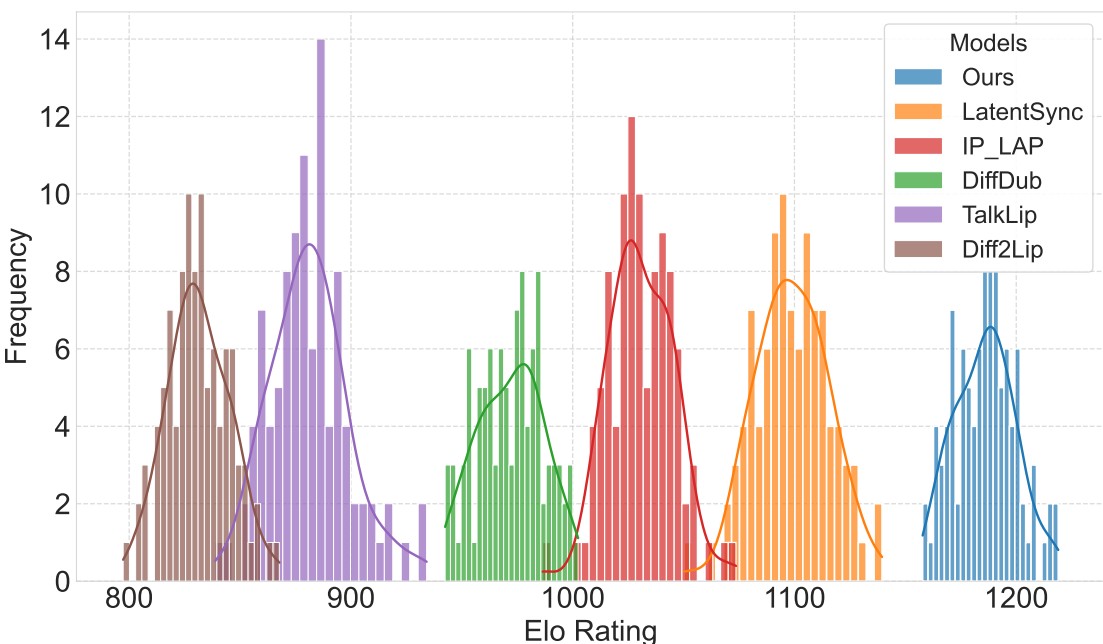

Figure 19: Distribution of Elo ratings across all evaluated models. This histogram illustrates the spread of Elo scores, highlighting performance gaps or clustering amongst different models.

# H   Additional Results

## H.1   Additional Baselines

In Section 5, we compare our method with five strong baselines. We also ran additional experiments against Wav2Lip Prajwal et al. (2020), due to its position as a foundational lip synchronization model, and VideoReTalking Cheng et al. (2022), as it also attempts to mitigate expression leakage. We present the results in Table 10. We observe that while VideoReTalking reduces leakage more effectively than Wav2Lip, its performance on cross-driving synchronization is still poor.

| Method | CMMD ↓ | TOPIQ ↑ | VL ↑ | FVD ↓ | LipScore ↑ | LipLeak ↓ |
|---|---|---|---|---|---|---|
| **Reconstruction** | | | | | | |
| VideoReTalking Cheng et al. (2022) | 0.263 | 0.45 | 29.28 | 536.12 | 0.45 | - |
| Wav2Lip Prajwal et al. (2020) | 0.201 | 0.44 | 27.59 | 506.41 | **0.48** | - |
| KeySync | **0.064** | **0.58** | **70.32** | **191.21** | 0.46 | - |
| **Cross-synchronization** | | | | | | |
| VideoReTalking Cheng et al. (2022) | 0.329 | 0.38 | 13.03 | 507.85 | 0.26 | 0.42 |
| Wav2Lip Prajwal et al. (2020) | 0.205 | 0.45 | 27.70 | 562.63 | 0.22 | 0.71 |
| KeySync | **0.070** | **0.58** | **73.04** | **206.32** | **0.48** | **0.22** |

Table 10: Additional quantitative comparison.

## H.2   Additional Synchronisation Metrics

While not included in our main comparison (Table 1) due to their known flaws, we present additional results using the Lip-Sync Error Confidence (LSE-C) and Lip-Sync Error Distance (LSE-D) metrics from

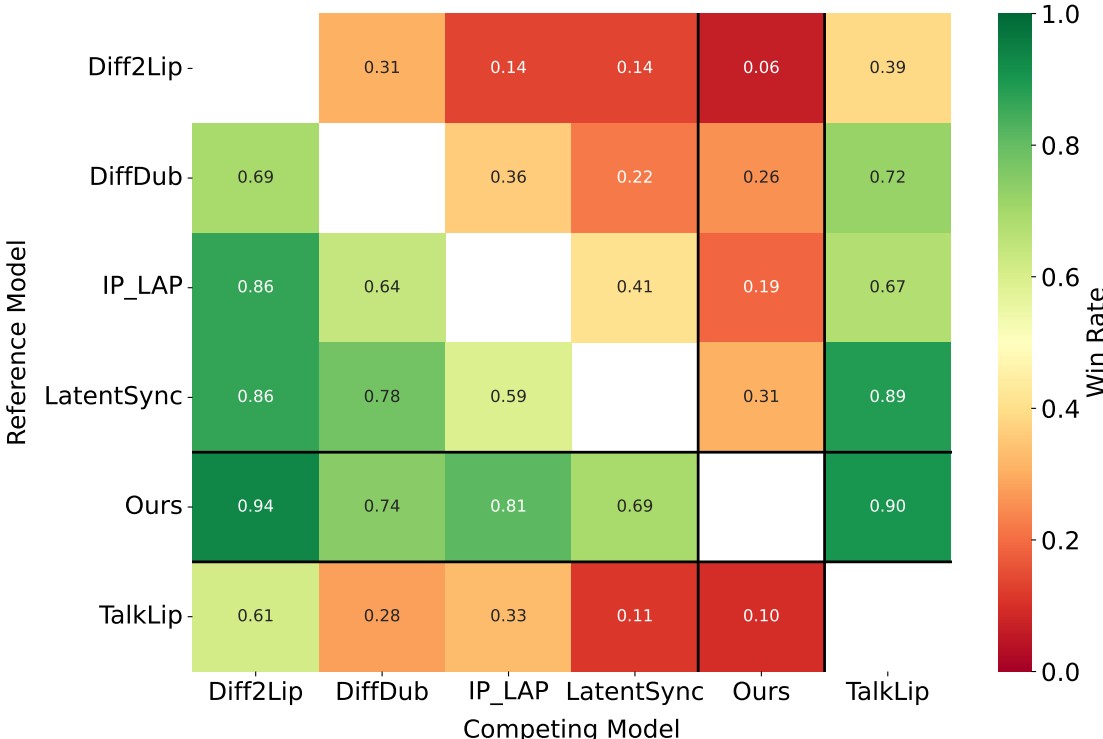

Figure 20: Win rate matrix for pairwise model comparisons. Each cell represents the proportion of matchups where one model outperforms another, offering insight into head-to-head performance.

SyncNet Chung & Zisserman (2016). Despite their limitations, these metrics remain widely used. The results for all baselines are shown in Table 11.

| Metric | DiffDub | IP-LAP | Diff2Lip | TalkLip | LatentSync | VideoReTalking | Wav2Lip | KeySync |
|---|---|---|---|---|---|---|---|---|
| LSE-D $\downarrow$ | 14.59 | 9.78 | 7.44 | 9.52 | 7.66 | 9.47 | 8.04 | **7.31** |
| LSE-C $\uparrow$ | 0.67 | 4.17 | 7.10 | 4.85 | 7.35 | 5.79 | 6.55 | **7.88** |

Table 11: Additional quantitative metrics using SyncNet.

# I   Additional Ablations

**Guidance**   Guidance plays a crucial role in the performance of diffusion models Dhariwal & Nichol (2021); Ho et al. (2020). In our case, we use a modified version of Classifier-Free Guidance (CFG) Ho (2022), which applies separate scaling factors to the audio and identity conditions. Specifically, our guidance function is defined as follows:

$$z = z_\emptyset + w_{\text{id}} \cdot (z_{\text{id}} - z_\emptyset) + w_{\text{aud}} \cdot (z_{\text{id \& aud}} - z_{\text{id}}), \tag{9}$$

where:

- $w_{\text{aud}}$ and $w_{\text{id}}$ are the guidance scales for audio and identity, respectively.

- $z_\emptyset$ represents the model output when all conditions are set to 0.

- $z_{\text{id}}$ is the output when only the identity condition is applied.

- $z_{\text{id \& aud}}$ is the output when both audio and identity conditions are applied.

By separating the audio and identity guidance conditions, we enable more control over the generated videos, ultimately leading to improved performance. Experimentally, we found that setting $w_{\text{aud}} = 5$ and $w_{\text{id}} = 2$ yields the best results. This configuration achieves a 29.73 % improvement in LipScore, significantly enhancing lip synchronization accuracy. While this comes at a 14.75 % increase in CMMD and a minor 2.80 % increase in FVD, the overall perceptual quality remains strong, making this trade-off highly beneficial for generating realistic and synchronized videos. We summarize these results in Table 12, demonstrating the effectiveness of our approach compared to standard CFG.

| Guidance | CMMD ↓ | FVD ↓ | LipScore ↑ |
|---|---|---|---|
| CFG | **0.061** | **200.71** | 0.37 |
| Ours ($w_{\text{aud}} = 5$, $w_{\text{id}} = 2$) | 0.070 | 206.32 | **0.48** |

Table 12: Guidance ablation in the cross-sync setting.

**Losses** We present an ablation on the impact of applying a pixel loss in addition to the diffusion loss in Table 13. Our findings indicate that adding a $L_2$ loss in pixel space leads to a slight improvement in image and video quality while maintaining the same level of lip synchronization. However, contrary to the findings in Bigata et al. (2025), we did not find that adding an additional LPIPS pixel loss benefits the model. Instead, it causes the mouth region to deviate too much from the rest of the image, as illustrated in Figure 21. This discrepancy arises because facial animation is a different task from lip synchronization, with the latter being more closely related to an inpainting task rather than full facial reconstruction.

| Loss | CMMD ↓ | FVD ↓ | LipScore ↑ |
|---|---|---|---|
| No pixel loss | 0.075 | 215.71 | **0.48** |
| $L_2$ | **0.070** | **206.32** | **0.48** |

Table 13: Pixel loss ablation in the cross-sync setting.

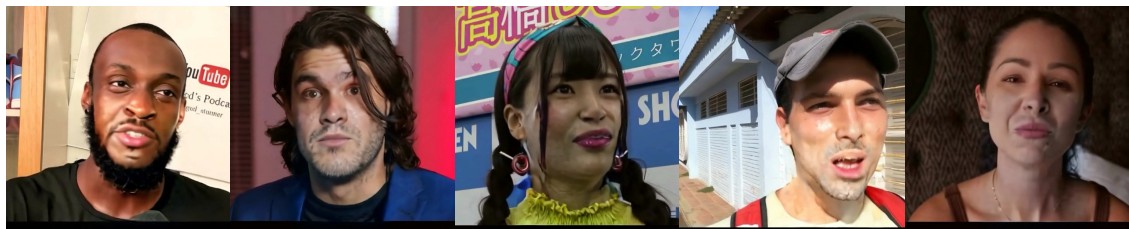

Figure 21: Examples of inconsistent mouth regions obtained by training with an additional LPIPS pixel loss.

## J Limitations

To assess the limitations of our approach, we construct a small dataset consisting of seven identities, where each individual recites the same two sentences at five different angles: 0°, 20°, 45°, 70°, and 90°, as illustrated in Figure 23. This setup allows us to systematically evaluate how the model performs under varying viewpoint conditions.

We present the results of TOPIQ Chen et al. (2024a) with respect to the angle in Figure 22. We use TOPIQ because it is a no-reference image quality metric that does not require a large ground-truth dataset for direct comparison, making it more practical than FID or FVD, which rely on reference distributions that may be skewed or incomplete across extreme angles. Additionally, unlike variance of Laplacian (VL), which only captures blurriness, TOPIQ provides a more comprehensive measure of perceptual quality degradation, including semantic distortions that become more pronounced at oblique head poses. The results indicate that all approaches exhibit performance degradation as the angle increases. This is a key limitation of our

model, which is also observed across baseline methods. This decline in performance can be attributed to the inherent biases in our training datasets, which predominantly contain frontal faces. As a result, the model struggles to infer occluded or unseen facial regions when presented with extreme head poses. One potential solution is to provide identity frames from multiple viewpoints during training, allowing the model to learn a more comprehensive facial representation. However, this would require extensive new data collection and further investigation, and is therefore left for future work.

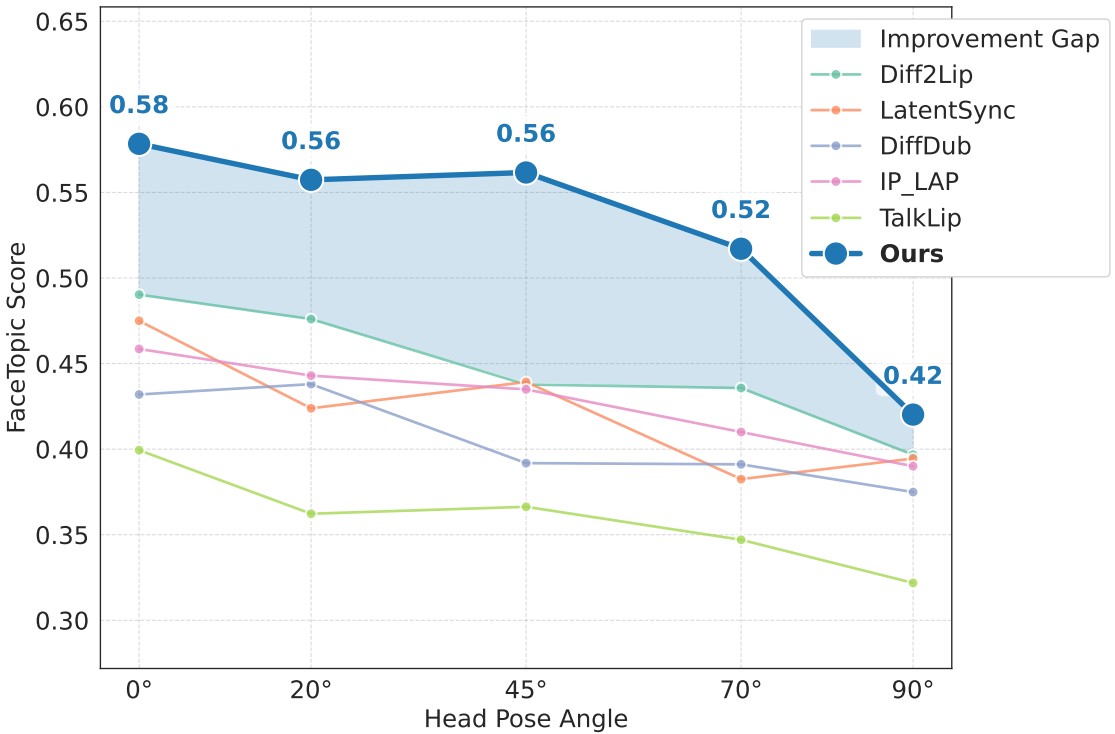

Figure 22: Impact of head pose on model performance.

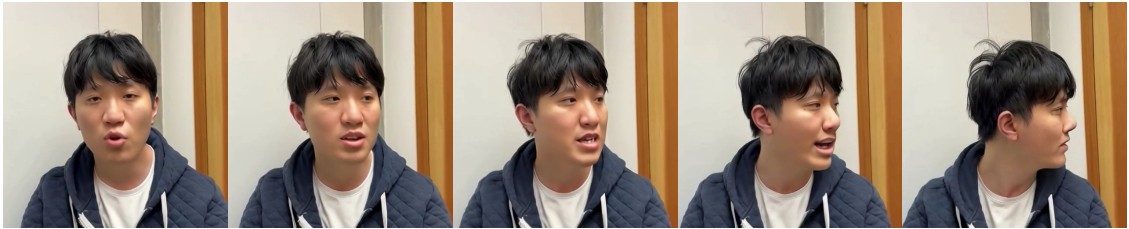

Figure 23: Examples of generated videos at different angles.

## K   Future work

The field of video generation is rapidly evolving, with a recent shift toward Diffusion Transformer Peebles & Xie (2023) (DiT) architectures (e.g., HunyuanVideo Kong et al. (2024), Wan Wan et al. (2025)) over the U-Net backbone used in SVD. We emphasize that KeySync is designed as a model-agnostic framework; our core contributions, specifically the leakage-proof latent masking and the two-stage inference logic, operate independently of the underlying denoising network. Consequently, our pipeline can be readily adapted to these emerging DiT backbones, where we anticipate the improved temporal attention mechanisms would further enhance the fidelity and motion coherence of our results.

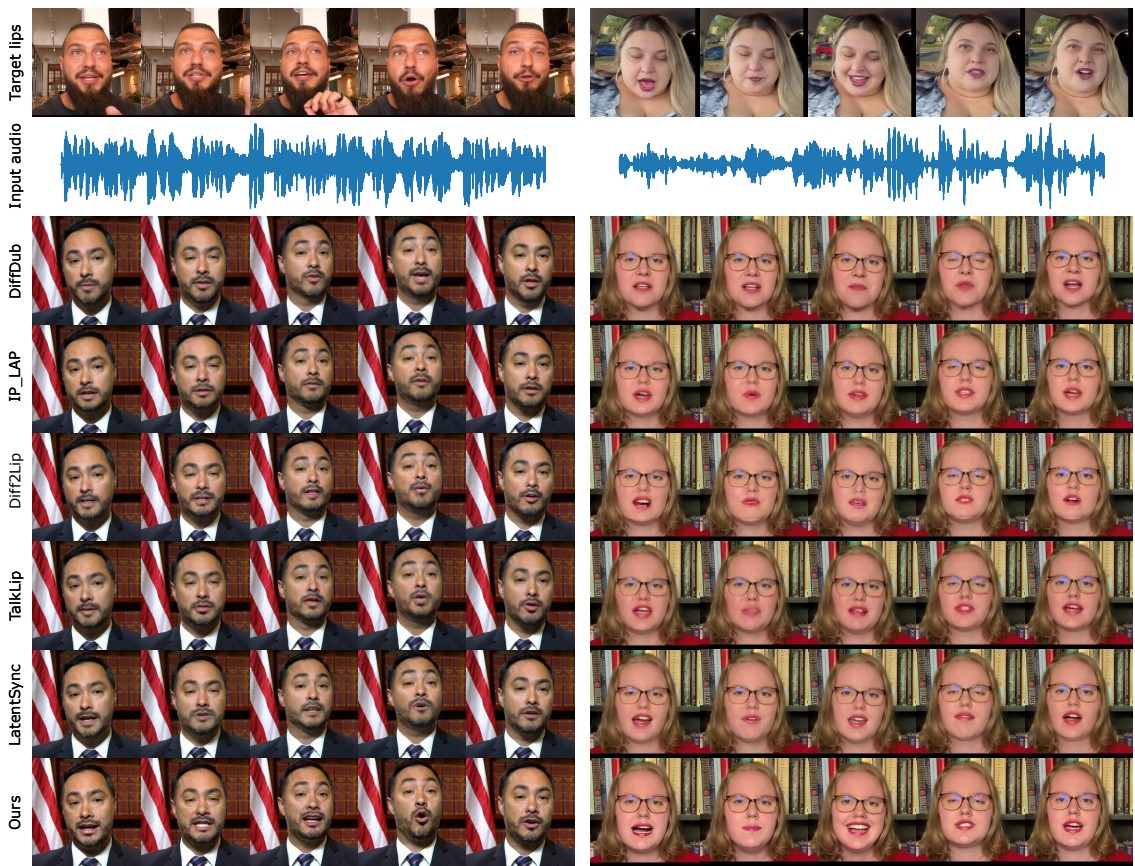

Figure 24: Additional qualitative comparison.

## L   Additional Qualitative Results

We present additional qualitative results in Figure 24. As reported in the main paper, our model demonstrates better alignment with the target lips while also achieving higher image quality compared to other methods. Additionally, we evaluate our model's ability to handle non-human faces in Figure 25. We find that KeySync produces plausible lip-synced animations, while competing models fail to accurately reconstruct mouth details, particularly in the first two identities, as they deviate significantly from typical human facial structures. This highlights our model's superior adaptability in handling out-of-distribution (OOD) scenarios.

To better assess the effectiveness of our approach, we provide a series of videos as part of the supplementary material. These videos are categorized as follows:

- **Side-by-side comparisons:** Showcasing our method against other approaches in both reconstruction and cross-sync settings.

- **Silent videos:** Highlighting expression leakage within the same video, demonstrating how different models handle silent audio.

- **Occlusion cases:** Also included in the same video, presenting situations where parts of the face are obstructed, illustrating the robustness of our approach.

- **Multilingual examples:** Evaluating the model's performance across different languages to assess generalization.

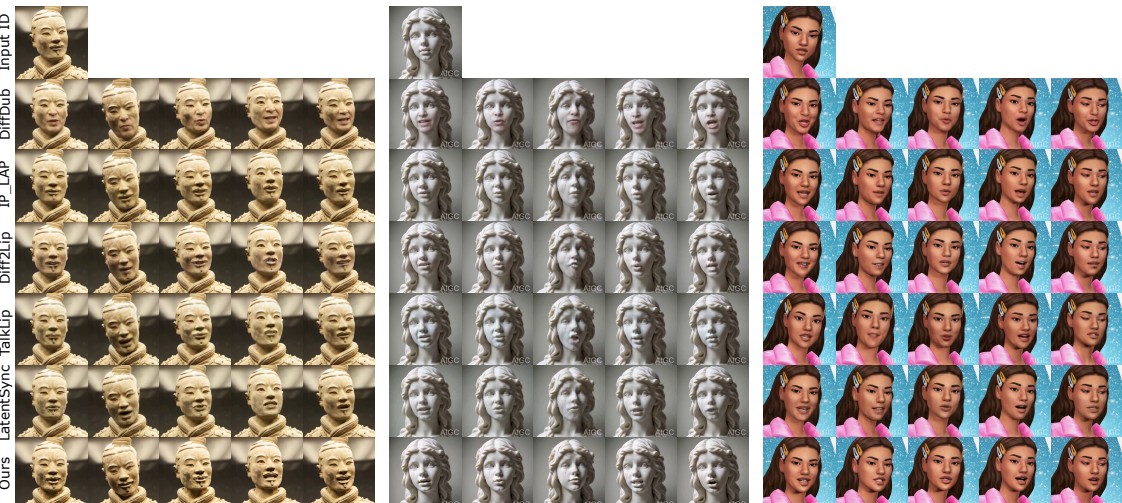

Figure 25: Qualitative comparison on non-human ids.

- **Out-of-distribution examples:** Testing our model on non-human identities, demonstrating its adaptability to non-human faces.

- **Examples at different angles:** Analyzing the model's performance under varying head poses, highlighting its ability to handle different viewpoints as well as its limitations.

- **Additional cross-sync videos:** Providing a more extensive evaluation of our model's cross-sync capabilities across various conditions.

These supplementary videos offer a comprehensive visual demonstration of our method's performance across a wide range of conditions.

## M    Ethical Considerations and Social Impact

**User study**    Our study includes a user evaluation where participants compare video outputs for lip synchronization, image quality, and coherence. All participants provided informed consent, and their responses were collected anonymously. No personally identifiable information or sensitive data were gathered, ensuring compliance with ethical research guidelines.

**Model**    Lip-sync generation offers numerous benefits, ranging from enhanced video dubbing to accessibility tools. However, we acknowledge the potential for misuse, particularly concerning deepfakes, misinformation, and identity fraud. Crucially, while our leakage-free synchronization significantly improves visual fidelity, it may inadvertently complicate detection methods that rely on identifying temporal inconsistencies. To mitigate these risks, we emphasise that this work is intended strictly for research purposes. Furthermore, as increased realism challenges existing safeguards, we advocate for the parallel development of detection frameworks that focus on the high-frequency artefacts and structural anomalies characteristic of diffusion-based generation, rather than relying solely on macroscopic visual errors.

**Datasets**    We rely on publicly available datasets that were originally collected and published by external researchers. We adhere to the terms and ethical guidelines set by the dataset creators.

