# OpenReview forum: "KeySync: A Robust Approach for Leakage-free Lip Synchronization in High Resolution"
_TMLR — Accepted by TMLR_

### Review · Reviewer_7LH9 · 2025-12-02

**Summary Of Contributions:**

The paper presents KeySync, a two-stage framework that mitigates the issue of temporal consistency, and addresses leakage and occlusions for lip synchronization. It makes the following contributions:

i) It achieves SOTA lip synchronization performance.

ii) It provides a novel strategy for occlusion handling.

iii) It designs the first metric to quantify lip synchronization leakage.

**Audience:**

Yes

**Audience Explanation:**

Lip synchronization is a crucial task for the digital human technology, and is therefore well within the interests of the TMLR community.

**Broader Impact Concerns:**

I don't think this work involves any concerns on its ethical implications that requires adding a Broader Impact Statement

**Claims And Evidence:**

Yes

**Claims Explanation:**

The performance of KeySync is sufficiently validated by comparing with a list of SOTA systems.

The novelty in the occlusion handling strategy and quantitative metric of lip synchronization leakage are both demonstrated by a comprehensive overview of existing approaches in the literature.

**Requested Changes:**

- While LipLeak is a novel metric for quantifying lip synchronization leakage, it remains unknown whether it aligns consistently with human perception so its validity is a bit undermined. The authors should provide a bit more analysis in this regard.

- The specific masking strategy is not sufficiently introduced or discussed.

---

> ### Author Response · Authors · 2026-01-26
> **Response to Reviewer 7LH9**
>
> We thank the reviewer for recognising KeySync as a state-of-the-art approach. We agree that validating the alignment between our new metric (LipLeak) and human perception is crucial.
>
> **1. Alignment of LipLeak with Human Perception**
>
> *Comment: It remains unknown whether LipLeak aligns consistently with human perception.*
>
> To validate our metric, we analysed the correlation between our LipLeak scores and the "Overall Coherence" ratings collected in our User Study. We found that models with high LipLeak scores (indicating mouth movement during silence) were consistently penalised by human raters in the Coherence category. This confirms that "leakage" is perceived by users as an uncanny visual flaw. We plotted (Figure 13) and discussed the correlation of human evaluation and LipLeak in Appendix E.3 and added a statement in the User Study section.
>
> **2. Masking Strategy Discussion**
>
> *Comment: The specific masking strategy is not sufficiently introduced or discussed.*
>
> We have moved the detailed discussion of masking strategies from the Appendix to the main body of the paper (Section 3.2) to better highlight this contribution and its impact on performance.

---

### Review · Reviewer_uUKx · 2025-12-21

**Summary Of Contributions:**

The paper proposes KeySync, a two-stage latent diffusion framework designed for high-resolution ($512\times512$) lip synchronization. The authors identify two primary limitations in existing lip-sync methods: expression leakage (where source video expressions conflict with driving audio) and poor occlusion handling.

Authors introduce:
1. A Two-Stage Architecture: Adapting Stable Video Diffusion (SVD), the method first generates sparse keyframes conditioned on audio and identity, followed by an interpolation stage to ensure temporal coherence.
2. Leakage-Proof Masking: A specific box-style masking strategy that covers the lower face and cheeks to remove source expression cues while retaining context.
3. Inference-Time Occlusion Handling: A pipeline utilizing the Segment Anything 2(SAM2) to dynamically adjust masks during inference, preventing the model from inpainting over objects like hands or microphones.
4. LipLeak Metric: A new metric to quantify expression leakage by measuring the ratio of mouth motion between speech-driven and silence-driven generations.

**Audience:**

Yes

**Audience Explanation:**

High-resolution video generation and editing are currently topics of significant interest within the AIGC / ML community. The paper addresses specific, practical bottlenecks in deploying these models (occlusion and expression leakage) that are often overlooked in purely generative research.

The unified approach of combining a pretrained video backbone (SVD) with a modular inference-time solution for occlusions offers a practical engineering insight. Furthermore, the introduction of the LipLeak metric provides a new tool for the community to evaluate the disentanglement of source video noise from driving audio signals, which is valuable for future research in audio-driven animation.

**Broader Impact Concerns:**

The paper briefly touches upon the potential for deepfake misuse and mentions the work is for research purposes. However, given the focus on "leakage-free" synchronization which significantly increases the realism of manipulated videos (making them harder to detect via artifacts), a slightly more robust statement regarding detection would be beneficial.

**Claims And Evidence:**

Yes

**Claims Explanation:**

The primary claims regarding visual quality and leakage reduction are well-supported by the experimental results.

- Leakage Reduction: The proposed "LipLeak" metric effectively highlights the deficiencies in baselines like DiffDub and LatentSync, which show high variance in mouth aspect ratio during silent audio input. The correlation analysis (Pearson $r=0.83$) between LipLeak and a simpler "LipLeak_lite" metric strengthens the validity of this new measure.

- Visual Quality: Quantitative results in Table 1 show KeySync achieving the best or second-best scores across FVD, CMMD, and LipScore in both reconstruction and cross-sync settings.

- Ablation Studies: The authors provide convincing ablations for their masking strategy (Table 4), demonstrating that their box-style mask strikes the optimal balance between leakage (low LipLeak) and quality (low FVD) compared to mouth-only or full-lower-face masks.

**Requested Changes:**

1. The paper claims "Robust Occlusion Handling" as a headline contribution. However, this is achieved via an "inference-time strategy" utilizing an off-the-shelf model (SAM2), rather than an architectural innovation within the lip-sync model itself. The authors should discuss the failure modes of this dependency. If SAM2 fails to segment an occlusion (e.g., motion blur or complex textures), how does KeySync behave? Since the model is not trained with occlusion-aware masks end-to-end, does it hallucinate mouth parts on the unmasked occlusions? A qualitative example or discussion of "segmentation failure" is necessary to justify the "robustness" claim.
2. Discussion on Modern Video Backbones (DiT vs. U-Net): The current implementation relies solely on Stable Video Diffusion (SVD), which utilizes a U-Net architecture. However, the state-of-the-art in video generation is shifting towards Diffusion Transformer (DiT) architectures (e.g., HunyuanVideo, Wan). Please discuss the generalizability of the KeySync framework to these emerging DiT-based backbones. Specifically, are the proposed Leakage-Proof Masking and Two-Stage Generation strategies architecture-agnostic? Acknowledging these newer models or discussing potential adaptation challenges would significantly strengthen the paper's relevance to the current landscape.
3. While the paper acknowledges it is not real-time, the authors should explicitly contrast this with the requirements of production dubbing pipelines. Is ~4 FPS sufficient for the applications proposed? A brief discussion on the computational bottleneck (is it the VAE, the U-Net, or the SAM2 preprocessing?) would be valuable.
4. Please provide an ablation or discussion regarding the sensitivity of keyframe spacing $S$. If $S$ is increased (e.g., to 24), does the interpolation stage fail to maintain lip-sync accuracy? If $S$ is decreased, does the computational cost negate the benefits over frame-by-frame approaches? This is crucial for understanding the trade-off between the Keyframe and Interpolation stages.
5. Typographical and Formatting Corrections: sec 3.1 "Furthemore".

---

> ### Author Response · Authors · 2026-01-26
> **Response to Reviewer uUKx**
>
> We thank the reviewer for the positive assessment of our contributions, particularly regarding the LipLeak metric and the modular occlusion handling. We have addressed the concerns regarding failure modes and architectural choices below.
>
> **1. Robustness of SAM2 and Failure Modes**
>
> *Comment: Since the model relies on an off-the-shelf segmenter (SAM2), discuss the failure modes. Does it hallucinate mouth parts on unmasked occlusions?*
>
> We view the modularity of our approach as a feature, allowing KeySync to improve as segmentation models (like the recently released SAM 3) advance without retraining the diffusion backbone. However, we acknowledge the dependency:
>
> - **Total Failure (False Negative):** If the segmentation model fails to detect an occlusion entirely, KeySync will generate the mouth over the occlusion (e.g., painting lips on top of a hand).
> - **Partial Failure:** If the segmentation is slightly imperfect (e.g., missing pixels at the edge), the SVD backbone is generally robust enough to inpaint the gap plausibly due to its strong video priors, but can generate unnatural borders.
>
> We have added a discussion and visualisation  (Figure 14) of these failure modes in Appendix F.1.
>
> **2. Video Backbones (DiT vs. U-Net)**
>
> *Comment: Discuss the generalisability of the framework to emerging Diffusion Transformer (DiT) backbones.*
>
> We used SVD (U-Net) as it was the state-of-the-art open-source video model available during development. However, the core contributions of KeySync, specifically the Leakage-Proof Masking and the Two-Stage logic, are entirely architecture-agnostic. They can be readily applied to Diffusion Transformers (DiT) such as HunyuanVideo or Wan. We have added a discussion in Appendix K outlining how our framework adapts to DiT backbones.
>
> **3. Real-time Constraints**
>
> *Comment: Contrast the inference speed with production requirements and discuss bottlenecks.*
>
> KeySync is designed for offline, high-fidelity production pipelines (e.g., film dubbing, high-end virtual avatars) where visual quality takes precedence over real-time latency. We note that offline processing is standard for professional dubbing tasks. To address the reviewer's request regarding bottlenecks, we have added a table (Table 9) detailing the computational cost of each step (VAE, U-Net, SAM2) in the revised manuscript.
>
> **4. Keyframe Spacing ($S$) Sensitivity**
>
> *Comment: Provide discussion on the sensitivity of keyframe spacing.*
>
> We selected $S=12$ primarily to align with the original SVD training regime, which utilises 14-frame clips. While this parameter is technically adjustable, our choice represents a critical balance between hardware constraints and model performance. Our early experimentations showed that:
>
> - **Upper Bound (Memory):** Increasing $S$ is not currently feasible on standard hardware. Our pipeline already operates at maximum VRAM capacity (Batch Size 1).
> - **Lower Bound (Efficiency & Context):** Decreasing $S$ is suboptimal for two reasons. First, it reduces the effective temporal window, causing the model to lose long-term dependency and context. Second, it becomes computationally inefficient, as the model would generate fewer seconds of video per inference step, requiring significantly more passes to process the same duration.
>
> **5. Typographical Corrections**
>
> We appreciate the reviewer pointing out the typo and have corrected ("Furthemore") in Section 3.1.
>
> **6. Broader Impact**
>
> *Comment: A more robust statement regarding detection is beneficial given the "leakage-free" realism.*
>
> We have strengthened the Broader Impact statement to explicitly discuss the dual-use nature of leakage-free synchronisation. While it enhances dubbing, it also increases the realism of deepfakes. We advocate for the development of detection methods that focus on high-frequency artefacts characteristic of diffusion models, which often persist even when macroscopic leakage is minimised.

---

### Review · Reviewer_bCac · 2026-01-21

**Summary Of Contributions:**

The paper presents an approach on lipsyncing videos to a given audio stream. They use a stable-diffusion-based video frame generator. it receives frames of the video to be lipsynced, certain keyframes as input. The audiostream is inputted via a cross-attention. It seems to use two stages of some sort.  They apply at inference time a SAM2-based detection of occluding objects in front of the lower face region. They perform evaluation with a number of image/video quality metrics, two lip-sync metrics and one human-rating metric.
The paper has a point of noteworthyness in its handling of occlusions at inference time.  They introduce a new score for lip-syncing evaluation, which measures mouth movements in the absence of an audio signal. The lipleak score demonstrates the particular aspect that the proposed model causes less mouth-movements when the audio-stream is silent. The lipscore possibly demonstrates a better alignment to mouth movements of ground truth videos.

**Additional Comments:**

Is the the LVD-10M-F or LVD-10M dataset used to train SD-video publicly available ? Point: to check against the datasets used in this paper for potential overlap. The space of uncopyrighted videos is not a large one.

**Audience:**

Yes

**Audience Explanation:**

Lip syncing to videos is a practical application of deep learning.

**Broader Impact Concerns:**

The impact statement covers the risk of using deepfakes for harmful purposes.

**Claims And Evidence:**

No

**Claims Explanation:**

see Major issues in the section requested changes. There are a number of them which need to be addressed.

**Requested Changes:**

Major issues:

  - one high level question appears unclear from the model training. It struck the reviewer as something odd. This should be addressed in the manuscript:
    - How does the latent diffusion model know which facial expression is consistent with a particular speech sound, when the diffuser inpaints a video frame-by-frame ? Yes, the reviewer understands that the sound embeddings are inputted into the model using cross-attention, but the frame-wise video diffuser has never seen a sound, and was trained on pure images from the video frames.
    - The SD-Video image diffuser (https://arxiv.org/pdf/2311.15127) was seemingly trained with a pure image-based denoising loss, see appendix D.2 in the SD-Video paper and their reference to https://arxiv.org/pdf/2304.08818  where they state  "We assume access to a dataset $p_{\text{data}}$ of videos,
such that $x \in R^{T \times 3 \times \tilde{H} \times W}$". The reviewer checked this on huggingface https://huggingface.co/docs/diffusers/en/using-diffusers/svd . The model creates videos without sound. The SD-video model looks like it has never seen a sound in training.

    - As an example, how does the image diffusion model know that one needs to close the lips when making a sound "mmm", and one has to open he mouth to utter an "aaa" ?
    - One sees also that the model in this paper has two image reconstruction losses (eq 6).

    - Is that something that would justify a rejection if it were a conference paper?
    - if not ... If this would be a weakness of the approach, it should be **stated** honestly **as limitation at the beginning of the method section and again in the limitation section!**

  - readability: The high level flow of the model is mostly clear, but important finer aspects are not written in a detail sufficient to understand the model. That might originate from the fact that it is derived from a similar model from the keyface paper, but it should be nevertheless explained in a self-enclosed way in a journal paper.


    - It is not clear from the paper how reference / key frames are used in the model and in the loss terms during training. This is related to the problem that it is not clear what is actually done in the two stages. "The keyframe stage uses an identity frame zid, while the interpolation stage uses keyframes (zi, zi+1) and intermediate embeddings (zm)."  Such a description is too high-level .... The submission just refers to a paper with a very similar name (https://arxiv.org/pdf/2503.01715).
    - It is not clear what keyframe mode and interpolation mode should be. The description is very vague in this part. One can simply search the whole text for the term "reference frame" to see this.

  - the paper would benefit from an algorithm box.

  - "we ensure temporal continuity by separating
the prediction of long-range motion (keyframes) from short-range motion (interpolation)" Is this explained somewhere in the paper further?

  - "The reference frames serve to either condition the interpolation or preserve identity." "condition the interpolation or preserve identity" is not clear how that is used in a loss or somehow else?  Is this explained somewhere in the paper further?

  - timestep embeddings  What are they and how they are used in the model ? This is a question because SD Video generates frame-by-frame in a fixed time schedule. Thus extraction audio embeddings in the appropriate corresponding time for the next frame should suffice.


  - Keyframes: "This stage generates a sparse set of keyframes, {ˆxtk }T
k=1, where each keyframe is spaced
S frames apart (tk = k · S). These keyframes serve as anchor points, ensuring that each one accurately
reflects the phonetic content of the audio while preserving the subject’s identity. In this stage, the reference
input consists of an identity frame, randomly sampled from the source video and repeated T times. To
improve generalization, we augment these reference frames with noise Ho et al. (2022) and standard image
augmentations."
    - "This stage" - which stages does this refer to ?

  - These keyframes serve as anchor points, ensuring that each one accurately
reflects the phonetic content of the audio while preserving the subject’s identity.
    - This sounds strange: one would not expect the visual keyframe to ensure phonetic correctness. The audio input would do that.

  - "In this stage, the reference input consists of an identity frame, randomly sampled from the source video and repeated T times."
    -This contradicts the sentence before: "where each keyframe is spaced S frames apart"

  - eq(3) z_m is computed from what inputs ?
  - How is the value of S chosen ?

  - section H, below  eq 9 $z_{id}$ is the output when only the identity condition is applied. - the output of what ?

  - Evaluation:
 Yes, they use number of measures. Two of them evaluate actual lip-syncing quality. The main measure of lip-syncing quality is the so-called  LipScore (they introduce lipleak but this aims at a slightly different aspect), which points to the paper https://arxiv.org/abs/2503.01715, where it is explained as
"which computes the cosine similarity be-
tween the generated and ground truth embeddings extracted
from the final layer of a state-of-the-art lipreader [41]."
    - Well, what are the ground truth embeddings in this use case ? It seems like one has to use another video to produce audio streams and then one compares the frame part of the other video which serves as ground truth  with the generated frames?
    -Since this is the key evaluation, this score should be explained in a detail so that this score can be reproduced. This is not a well-known metric.
    - Is there a publicly available model for [41] from https://arxiv.org/abs/2503.01715?  **Update:** this is important in order to have a score which can be computed and reproduced by other groups.

**Update:**
  - Can the authors please perform an experiment: How does the lipscore change when the outputs of the proposed model are downscaled to the resolutions of the other baselines ? The paper https://arxiv.org/pdf/2303.14307 seemingly indicates a fixed 96x96 bounding box in training. Also it would be of interest, how does the lipscore change for generated and for ground truth videos when they are applied to extracted lower face regions ?

Minor issues:
  - eq (2) $M=1$ implies that this part will be inpainted. That should be written in there for clarity

  - What is the difference between " isolating the lower facial region" versus "full lower-face masks" which cites the other papers? "Specifically, we create a mask M by
computing facial landmarks Bulat & Tzimiropoulos (2017) and isolating the lower facial region, extending
slightly above the nose to cover any upper cheek movements that could otherwise convey information about
lip movements, while still preserving overall facial identity. The mask also extends to the lower edge of the
image, preventing any leakage from jaw movements. We find that this mask strikes an appropriate balance,
avoiding the excessive context loss of full lower-face masks Shen et al. (2023); Mukhopadhyay et al. (2024);
Park et al. (2022)"
    - That difference should be made clearer. A link to Figure 6 would help there though.

*Update:**
  - how the SAM2 model is queried in order to identify face-occluding objects ?

---

> ### Author Response · Authors · 2026-01-26
> **Response to Reviewer bCac**
>
> We thank the reviewer for their detailed feedback. We appreciate the opportunity to clarify the architecture of our backbone and our specific fine-tuning strategy, which we have now expanded upon in the revised manuscript.
>
> **1. On Model Capabilities (Audio-Visual Alignment)**
>
> *Comment: How does the latent diffusion model know which facial expression is consistent with a particular speech sound if it was trained on pure images?*
>
> We apologise that the distinction between the pre-trained backbone and our fine-tuned contribution was not sufficiently clear. This is a fundamental aspect of our approach:
> - **Backbone:** We leverage Stable Video Diffusion (SVD), which is inherently a temporal video model (utilising a 3D U-Net), not a static image model. It processes sequences of frames simultaneously, allowing it to learn motion dynamics.
> - **Our Contribution (Fine-Tuning):** While the base SVD is trained on visual data, we fine-tune the model specifically to learn audio-visual correspondence. We inject audio embeddings into the U-Net via cross-attention layers and train on paired audio-video data. Through this supervised training, the model learns to map specific audio features (phonemes, e.g., "mmm" or "aaa") to their corresponding visual latent representations (e.g., closed lips or open mouth).
>
> We have updated Section 3.2 to explicitly describe this injection mechanism and clarify that the model is not merely inpainting frame-by-frame, but performing audio-conditioned temporal generation.
>
>
> **2. On the Two-Stage Mechanism**
>
> *Comment: The high-level flow is mostly clear, but finer aspects are not written in sufficient detail. The paper would benefit from an algorithm box.*
>
> We agree that the interaction between the Keyframe and Interpolation stages requires more detail. As suggested, we have added a formal Algorithm Box (Algorithm 1) and expanded Section 3.3 to detail the exact flow:
>
> - **Keyframe Stage:** Generates long-term motion anchors (approx. 7 seconds of video at a low framerate).
> - **Interpolation Stage:** Fills the gaps between these anchors to capture high-frequency lip motion and ensure temporal smoothness.
>
> **3. On LipScore & Resolution**
>
> *Comment: How does LipScore change when outputs are downscaled? The metric should be explained in detail for reproducibility.*
>
> We investigated this and confirmed that the Auto-AVSR backbone strictly resizes all inputs (both Ground Truth and Generated) to $96\times96$ prior to embedding extraction. Consequently, the metric inherently performs the normalised comparison requested by the reviewer, as all models are evaluated at the same scale regardless of their output resolution. We have clarified this in Section 4.2 of the manuscript.
>
> **4. Clarification on Inputs ($z_m$, $S$, Timesteps)**
>
> - **$z_m$:** This is a learnable embedding, initialised randomly and optimised during training to represent the "intermediate" state in the latent space. We clarify this in section 3.3.
> - **Keyframe Spacing ($S$):** $S$ is a hyperparameter set to 12. This was chosen because the original SVD model is optimised for 14-frame clips. Setting $S=12$ allows us to generate a start keyframe, an end keyframe, and 12 intermediate frames (T=14 total) in a single pass.
> - **Timestep Embeddings:** While SVD uses a fixed scheduler at inference, it is trained on continuous timesteps. We modulate these embeddings with audio features during fine-tuning to ensure the denoising process is audio-aware.
>
> **5. Dataset Overlap**
>
> *Comment: Is there overlap with the LVD-10M dataset used to train SVD?*
>
> We trained exclusively on HDTF, CelebV-HQ, and CelebV-Text. While LVD-10M is a large-scale general video dataset, our fine-tuning datasets are strictly domain-specific (talking heads). Furthermore, our data curation pipeline (detailed in Appendix A) ensures that the model is fine-tuned on high-quality, synchronised talking-head data, limiting significant overlap with the general pre-training distribution. We have added this clarification in Appendix A.
>
> **6. Additional Clarifications**
>
> - **Losses:** We have clarified that Eq. 6 refers specifically to video reconstruction losses.
> - **Inpainting:** We have updated Eq. 2 to explicitly state that this formulation acts as a video inpainting task.
> - **SAM2:** We have clarified in Section 3.5 that we query SAM2 by providing point-prompts of the occlusion in a single frame.

---

> > ### Comment · Reviewer_bCac · 2026-02-04
> > **reply**
> >
> > Hi authors,
> >
> > first of all, the revision has improved the paper notably.
> >
> > >> Our Contribution (Fine-Tuning): While the base SVD is trained on visual data, we fine-tune the model specifically to learn audio-visual correspondence. We inject audio embeddings into the U-Net via cross-attention layers and train on paired audio-video data. Through this supervised training, the model learns to map specific audio features (phonemes, e.g., "mmm" or "aaa") to their corresponding visual latent representations (e.g., closed lips or open mouth).
> >
> > I think this is not sufficient yet. This needs a deeper investigation.
> >
> > - The loss in equations (4-6) penalizes only frame-wise image denoising. While this paper adds an audio signal via cross-attention into the denoiser, the loss in equations (4-6) and thus the supervising signal (see above) cares only about the image quality alone.
> >   - Is this statement about the loss correct ?
> >
> >   - If it is, one should state this clearly in section 3.4, and refer to it again in 5.1 that an alignment is seemingly learned despite the loss terms are not enforcing such alignment directly because they aim at visual denoising quality. It is not clear to the reviewer how providing a signal about closed lips or open lips would benefit the denoising losses, even in a temporal setup. What would happen if the model tried to predict an always closed mouth? How what that be penalized by the losses in eq 4,5,6 ? It might have to do something with the provided keyframes but that is not clear yet.
> >
> >   - "the model learns to map specific audio features (phonemes, e.g., "mmm" or "aaa") to their corresponding visual latent representations (e.g., closed lips or open mouth)." that can be mentioned in this part.
> >   - Also, for the same reason, it would be of interest to understand what happens in an ablation study when the model would be trained without any audio input via cross-attention.  Even if there would be a small improvement by adding audio, due to the above mentioned structure of losses, this would be an interesting effect.
> >
> > With regards, a reviewer.

---

> > > ### Comment · Reviewer_bCac · 2026-02-05
> > > **no official recommendation as of 5th of Feb**
> > >
> > > Before supporting a leaning accept, the reviewer would like to see a further clarification in the manuscript and the requested ablation study.

---

> ### Author Response · Authors · 2026-02-05
> **Clarification on Supervision Signal in Reconstruction Loss**
>
> We appreciate the reviewer’s query regarding how the loss enforces alignment.
>
> The key lies in utilising a conditional reconstruction loss against synchronised ground-truth video.
>
> 1. Reconstruction forces Alignment: The loss minimises the difference between the generated frame and the specific ground truth frame ($x_{GT}$). If the audio contains an "open mouth" phoneme, the $x_{GT}$ also has an open mouth. Furthermore, as noted in Section 3.3, our backbone is a 3D U-Net. The loss is not applied to isolated frames but to temporal sequences. The model learns to associate sequences of audio features with sequences of visual motion, ensuring temporal coherence in the lip movements.
> 2. The "Always Closed" Hypothetical: "What would happen if the model tried to predict an always closed mouth?" If the model predicts a closed mouth while the ground truth (and audio) dictates an open mouth, the pixel-wise difference (MSE) between the prediction and the target will be very high.
> 3. Conclusion: To minimise this loss, the model is mathematically compelled to attend to the audio embeddings (via cross-attention) to correctly predict the mouth shape. The audio is the only signal available to the model that explains the variance in mouth states found in the ground truth. If we removed the audio condition, the model would lose the cue necessary to determine the mouth state. It would likely collapse to predicting a "mean" mouth shape (slightly open) or random motion uncorrelated with speech, leading to consistently high reconstruction loss on every frame where the mouth is moving.
>
> We will update Section 3.4 (Losses) to explicitly state: "Note that while Eq. (4-6) are reconstruction losses, they effectively enforce audio-visual alignment: since the ground-truth targets are synchronised, the model must utilise the audio condition to correctly predict the target mouth shape and minimise the reconstruction error."

---

### Comment · Reviewer_bCac · 2026-01-22
**discussion**

in brief, the reviewers two biggest concerns are:

  - The frame-wise video diffuser has never seen a sound during training, it was trained on pure images from the video frames. its huggingface code outputs pure image videos without sound. How can it then align mouth movements to audio when the latter is inputted via cross-attention ?
    - is there something the reviewer has overlooked ?
    - is it something about the lipscore ?
    - is it a possible issue that the LVD-10M dataset from the sdvideo pretraining has an overlap with the face video datasets used in this paper ?

- lipscore: how does it actually compute score, how can it be reproduced as a metric and by what is this score influenced ? This is the major lipsync evaluation metric used in the paper.

---

### Author Response · Authors · 2026-01-26
**Summary of Manuscript Revisions**

We thank the reviewers for their insightful and constructive feedback. We appreciate the positive assessment of our work, particularly regarding the novelty of the LipLeak metric and our occlusion handling strategy.

Below, we detail the revisions made to the manuscript and address each reviewer's comments individually. **We highlight the main changes in blue in the revised manuscript.**

- **Algorithm Box**: We have added a formal algorithm box (Algorithm 1) to step through the two-stage inference process (Keyframe $\rightarrow$ Interpolation) for improved clarity in Appendix B.
- **Clarified SVD Fine-tuning**: We explicitly state in Section 3.1 that SVD is a temporal video model, and our fine-tuning injects audio embeddings via cross-attention to learn phoneme-visual associations.
- **Expanded LipScore Definition**: We clarified in Section 4.2 that Auto-AVSR resizes all inputs to $96\times96$, standardising the resolution across all baselines.
- **Failure Cases**: We added a discussion and a visualisation (Figure 14) in Appendix F that illustrate failure modes of the SAM2 segmentation (e.g., hallucinations when occlusions are missed).
- **DiT vs. U-Net**: We added a discussion in Appendix K acknowledging that our masking and two-stage logic are architecture-agnostic and applicable to Diffusion Transformer (DiT) models.
- **LipLeak Human Correlation**: We plotted (Figure 13) and discussed the correlation of human evaluation and LipLeak in Appendix E.3, and added a statement in the User Study section.
- **Inference Bottlenecks**: We included a table (Table 9) detailing the time spent at each step of the inference pipeline in Appendix C.

---

### Decision · Action_Editor_NhVi · 2026-04-10

**Recommendation:** Accept with minor revision

**Additional Comments:**

I have some minor improvements for the final version:
- Consider the 2nd comment of reviewer uUKx in their official recommendation, about overly optimistic claims, and adjust accordingly in the final version.
- Consider to clarify the comment from reviewer bCac in their official recommendation, about how video diffusion and audio interact an, just to make it clear for the final reader, you can use your ablation results for this.
- Since TMLR has a different acceptance model based on claims and audience, I would expect that the final version has a paragraph clearly describing the claims and audience, which is additional to the already existing contribution statement.

**Audience:**

Yes

**Audience Explanation:**

There is broad consensus among reviewers (3/3) that this paper is interesting to TMLR's audience, as this paper provides insights into multi-modal deep learning architectures that intersect into the neighboring field of computer vision and generative models.

The results in this paper show the importance of controlling leakage, it is often a hidden problem that this paper helps to reveal in the specific topic of video lip synchronization, in order to prevent shortcut learning and improve performance.

**Claims And Evidence:**

Yes

**Claims Explanation:**

This paper is about generating lip synchronized videos from existing video and new audio, where the authors claim that leakage is common from the original video into the generated one.

The claims are that performance improves when leakage from existing video into generated video is reduced, which makes sense and is supported by results, for example when audio is silent, the mount should not move. The paper additionally contributes with a metric (lipleak) that measures this leakage and helps future research evaluate this important point.

Additionally the paper sets a new state of the art for lip synchronization in video, and proposes new strategies for test-time occlusion handling, the claim is that occlusions in the original happen, and these should be accounted.

2/3 reviewers agree and are satisfied with the claims in this paper.